# Molecular model of TFIIH recruitment to the transcription-coupled repair machinery

Tanmoy Paul[1,2], Chunli Yan [1,2], Jina Yu[1,2], Susan E. Tsutakawa [3], John A. Tainer [3,4], Dong Wang [5] & Ivaylo Ivanov [1,2] ✉

Transcription-coupled repair (TCR) is a vital nucleotide excision repair sub-pathway that removes DNA lesions from actively transcribed DNA strands. Binding of CSB to lesion-stalled RNA Polymerase II (Pol II) initiates TCR by triggering the recruitment of downstream repair factors. Yet it remains unknown how transcription factor IIH (TFIIH) is recruited to the intact TCR complex. Combining existing structural data with AlphaFold predictions, we build an integrative model of the initial TFIIH-bound TCR complex. We show how TFIIH can be first recruited in an open repair-inhibited conformation, which requires subsequent CAK module removal and conformational closure to process damaged DNA. In our model, CSB, CSA, UVSSA, elongation factor 1 (ELOF1), and specific Pol II and UVSSA-bound ubiquitin moieties come together to provide interaction interfaces needed for TFIIH recruitment. STK19 acts as a linchpin of the assembly, orienting the incoming TFIIH and bridging Pol II to core TCR factors and DNA. Molecular simulations of the TCR-associated CRL4$^{CSA}$ ubiquitin ligase complex unveil the interplay of segmental DDB1 flexibility, continuous Cullin4A flexibility, and the key role of ELOF1 for Pol II ubiquitination that enables TCR. Collectively, these findings elucidate the coordinated assembly of repair proteins in early TCR.

DNA lesions in the template strand of actively transcribed genes pose a major threat to gene expression and left unchecked, could result in genome-wide transcriptional arrest[1,2]. To counter this threat, cells have evolved elaborate DNA repair pathways. Specifically, the versatile nucleotide excision repair (NER) pathway removes from genomic DNA a vast array of bulky or helix distorting lesions caused by ultraviolet radiation, reactive oxygen species, environmental carcinogens, and chemotherapeutic agents such as cisplatin[3–6]. Moreover, NER's transcription-coupled repair sub-pathway (denoted as TCR or TC–NER) efficiently eliminates transcription-blocking DNA lesions[7–10]. TCR is key for genome integrity and, predictably, its functional impairment by mutations causes inherited genetic disorders—Cockayne syndrome (CS)[11–15] and UV-sensitive syndrome (UVSS)[16,17]. Phenotypes associated with CS and UVSS are markedly dissimilar. CS patients carry mutations in the CSB or CSA genes, leading to developmental abnormalities, severe progressive neurodegeneration, and premature ageing[12,18,19]. By contrast, UVSS patients carry mutations primarily in the UVSSA gene, which result in mild UV photosensitivity without neurological features[17,20]. Yet, a molecular basis for this striking divergence in clinical outcomes has not emerged. Lesion recognition in TCR involves stalling of elongating RNA polymerase II (Pol II or RNAPIIo) at sites of DNA damage[21,22]. Yet, Pol II stalling also shields the damaged strand from direct action by repair enzymes. Instead, TCR proceeds through a complex assembly mechanism, whereby core TCR proteins—Cockayne Syndrome B protein (CSB/ERCC6)[22], Cockayne syndrome protein A (CSA/ERCC8), and UV-sensitive syndrome protein (UVSSA)—sequentially engage stalled RNAPIIo, eventually triggering the recruitment of transcription factor IIH (TFIIH)[23–29]. TFIIH comprises

[1]Department of Chemistry, Georgia State University, Atlanta, GA, USA. [2]Center for Diagnostics and Therapeutics, Georgia State University, Atlanta, GA, USA. [3]Molecular Biophysics and Integrated Bioimaging, Lawrence Berkeley National Laboratory, Berkeley, CA, USA. [4]Department of Molecular and Cellular Oncology, The University of Texas MD Anderson Cancer Center, Houston, TX, USA. [5]Division of Pharmaceutical Sciences, Skaggs School of Pharmacy & Pharmaceutical Sciences, University of California, CA San Diego, USA. ✉e-mail: iivanov@gsu.edu

ten protein subunits—seven forming core TFIIH (XPD, XPB, p44, p34, p8, p62, and p52) and three forming the CAK subcomplex (MAT1, Cdk7, and Cyclin H)[30,31]. While CAK is key for TFIIH's function in transcription, its dissociation from core TFIIH is required for functional NER. Nonetheless, the CAK subunits associate with lesion-stalled RNAPIIo during early TCR[32,33]. The precise mechanistic role of this association is unclear. TFIIH harbors two translocase subunits XPB and XPD[34], which unwind DNA near the Pol II cleft and expose the DNA lesion. XPD then verifies the lesion by scanning the damage-containing DNA strand[34–36]. Lesion verification is followed by coordinated dual incision by two structure-specific nucleases, XPG and XPF/ERCC1[37–44], and the release of a 22–30-nucleotide DNA segment containing the lesion[45]. Subsequently, Polδ, RFC, and PCNA are loaded onto the ensuing gap substrate to initiate gap-filling DNA synthesis and restore the excised region. Lastly, DNA ligase seals the nicked DNA strand to complete the repair process. Thus, TCR has evolved a complex mechanism wherein stalled RNAPIIo serves as a platform for the recruitment of the core NER protein machinery.

As the centerpiece of this complex TCR machinery, TFIIH unwinds DNA to expose the lesion[9,25,46], is considered to cause Pol II backtracking and/or dissociation[10], and coordinates the dual incision process. This progression ultimately enables downstream repair proteins to access and repair the lesion. Yet, our understanding of how TFIIH is initially recruited and functions within the TCR repair machinery remains incomplete. Previous structural and biochemical work[33,47–49] posited a cooperative assembly mechanism of a TCR complex in which sequential recruitment of CSB, CSA, and UVSSA targets TFIIH to lesion-arrested RNAPIIo to initiate repair.

The CSB protein is a SWI2/SNF2 chromatin remodeler that engages RNAPIIo upstream of the transcription bubble and promotes forward Pol II translocation, facilitating transcriptional bypass of less bulky lesions. The CSA protein contains seven WD40 repeats that form a seven-bladed β-propeller, which is flanked on its opposing sides by CSB and UVSSA. CSA binding is mediated by two conserved interaction motifs – the CSA-interaction region (CIR) in UVSSA (residues 100-200) and the CSA-interacting motif (CIM) in CSB (residues 1385-1399)[33]. In addition to serving as a bridge between CSB and UVSSA, CSA[48] also functions as the substrate-recognition subunit of a DDB1–CUL4A–RBX1 ubiquitin ligase complex (CRL4$^{CSA}$)[49] that becomes transiently activated in response to UV irradiation[47,50,51]. Assisted by elongation factor ELOF1[52], CRL4$^{CSA}$ specifically ubiquitinates the Pol II Rpb1 subunit at position K1268[47,52]. In turn, Rpb1-K1268 ubiquitination stimulates TFIIH binding to stalled RNAPIIo via a transfer mechanism that is also dependent on UVSSA-K414 mono-ubiquitination[50]. Alternatively, failed TFIIH recruitment triggers CSB polyubiquitination by CRL4$^{CSA}$ and targets it for proteasomal degradation[53]. Moreover, the CSB protein harbors a UBD domain whose role in TCR remains to be established[54,55]. The UBD must recognize a ubiquitin moiety either on CSB itself or on another protein (e.g., Rpb1) but neither the ubiquitin partner position nor the mechanistic consequences of this interaction are known.

UVSSA is the key partner protein that recruits TFIIH to the expanding TCR assembly through a TFIIH-interaction region (TIR; residues 400-500)[33]. TIR harbors a conserved acidic patch motif (residues 408-412) that binds the pleckstrin-homology domain (PHD) of the TFIIH p62 subunit and is essential for TCR[56]. Intriguingly, the immediately adjacent K414 residue has been identified as the key mono-ubiquitination site in UVSSA in response to UV irradiation. Mutations that prevent the p62–UVSSA interaction via the PHD also impair TFIIH–RNAPIIo association and TCR. By contrast, a UVSSA-K414R mutant is TCR deficient while retaining normal TFIIH-p62 binding. Thus, UVSSA-K414 ubiquitination and the UVSSA-PHD interaction are both functionally independent and crucial for TCR. Their interplay within the complex repair mechanism remains to be established. Similarly, both Rpb1-K1268 and UVSSA-K414 ubiquitination are

independently required for TFIIH recruitment to lesion-arrested RNAPIIo[49,50]. However, the precise functional roles of the two distinct ubiquitination events in the repair mechanism and the molecular interactions leading to TFIIH recruitment remain unexplored. Finally, a new putative TCR protein, inactive serine threonine kinase 19 (STK19)[57,58], was identified by multi-omics screening and later shown to impact sensitivity to DNA damaging agents in CRISPR sensitivity screens. Originally misclassified as a serine threonine kinase, STK19 was later shown to have no detectable kinase activity[57,59]. While STK19 is known to impact cellular response to UV DNA damage, its precise functional role in TCR is unresolved.

Much of the recent progress in understanding TCR mechanisms has come from cryo-EM studies[22,47,48]. Yet, no structure of a TFIIH-bound TCR complex exists and many outstanding mechanistic questions remain unresolved. How is STK19, a DNA/RNA-binding protein critical for DNA damage repair[57,60–62], involved in the TCR mechanism? Do STK19 and elongation factor ELOF1 interact with the incoming TFIIH assembly and, if so, what are the relevant protein–protein interfaces? Once recruited, how does TFIIH act to unwind DNA? In what way could TFIIH recruitment downstream of the lesion backtrack or displace RNAPIIo and give access to downstream repair factors?

Here we have used integrative molecular modeling to synthesize existing structural data on TCR protein assemblies and AlphaFold-multimer[63–65] predictions to model the initial recruitment of TFIIH to the human TCR machinery. The new TCR–TFIIH models enable side-by-side comparisons to known structures of transcription pre-initiation complexes (PICs)[28,66–68], and global genome NER (GG-NER) complexes[69–72]. Importantly, our findings suggest clear functional roles for key constituent proteins (CSB, CSA, DDB1, Cullin4A, UVSSA, ELOF1, and STK19) in stabilizing the nascent TCR–TFIIH complex. The structural models were further used to initiate microsecond-timescale molecular dynamics (MD) simulations that elucidate the global motions of the TCR–CRL4$^{CSA}$ and TCR–TFIIH molecular machinery. We employ graph-theoretical algorithms to partition the TCR assemblies into dynamic communities and network analysis to reveal key aspects of TFIIH dynamics in TCR. Molecular simulations also shed light on the functional roles of the elongation factor ELOF1 and the CRL4$^{CSA}$ complex in promoting specific Pol II ubiquitination for TFIIH recruitment. Our integrative modeling provides a roadmap for future experiments to test the interplay between the structural disruption of TCR complexes by mutations and the etiology of genetic diseases such as CS and UVSS. Collectively, our results unveil how the TCR molecular machinery dynamically reshapes itself to achieve precise lesion removal and preserve genome integrity.

## Results

### Molecular architecture of the nascent TCR–TFIIH complex

While cryo-EM studies have shed light on the functional states of the TCR machinery immediately preceding TFIIH recruitment, no structures exist of TFIIH-bound TCR complexes. Core TFIIH is known to adopt two principal conformational states – 1) an open 'horseshoe' shaped arrangement of the seven TFIIH subunits as observed in the transcription preinitiation complex (PIC)[28,66]; or 2) a closed circular subunit arrangement characteristic of GG-NER[70,73]. To decide if the TCR–TFIIH complex is compatible with the open or circular form of TFIIH, we first created structure overlays of a recently determined TCR assembly (comprised of RNAPIIo, CSB, CSA, UVSSA, DDB1, and ELOF1)[47] with the human PIC[66] and a TFIIH–XPA–DNA complex[73] (Supplementary Fig. 1). We found that the XPD subunit in the circular TFIIH–XPA–DNA structure severely clashed with key TCR proteins, notably CSA and DDB1, making the two structures incompatible. By contrast, the open TFIIH conformer from the PIC could be accommodated with no clashes and provided highly complementary interfaces for TFIIH association (Fig. 1a, Supplementary Fig. 1). Thus, early TFIIH recruitment occurs in an open conformational state wherein the two TFIIH translocase subunits (XPB and XPD) are distal, and XPD is

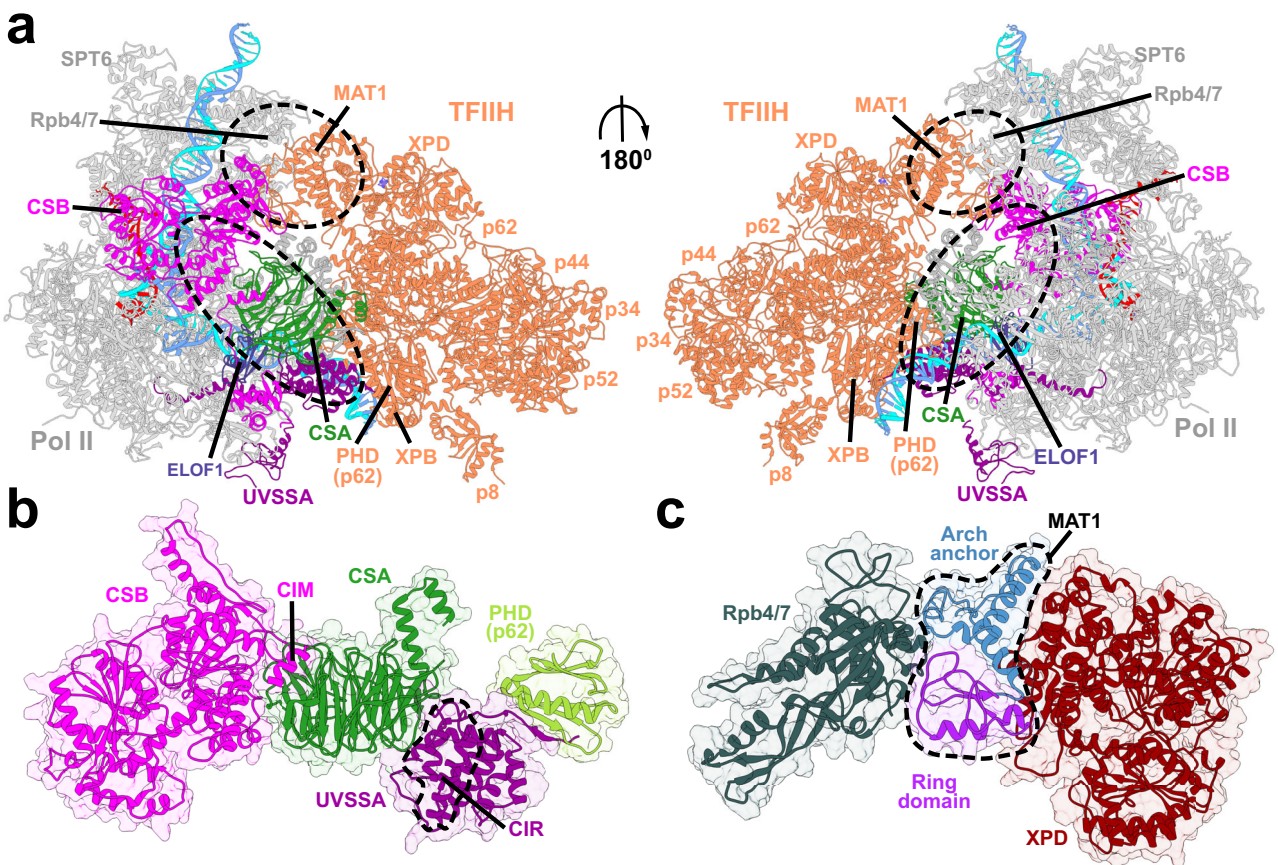

**Fig. 1 | Integrative model of the TCR–TFIIH complex reveals the overall structural organization of the assembly. a** Anterior and posterior views of the TCR–TFIIH complex. TFIIH is shown in orange, CSB in magenta, CSA in dark green, UVSSA in purple, and ELOF1 in dark blue. The principal structural modules bridging RNAPIIo and TFIIH in the complex are outlined by dashed lines. Zoomed-in views of (**b**), the CSB–CSA–VHS module and the UVSSA–p62 PHD interface; **c**, the MAT1–Rpb4/7–XPD interface. Conserved interaction motifs: the CSA Interacting Motif (CIM) of CSB and the CSA Interacting Region (CIR) of UVSSA are labeled. The PH domain is shown in light green. The Arch anchor and RING domains of MAT1 are shown in steel blue and purple, respectively.

expressly sequestered away from substrate DNA. Subsequent DNA unwinding activity of XPB, recruitment of XPA, and XPD repositioning transforms this initial 'inactive' complex into an 'active' repair-competent assembly. In creating the integrative TCR–TFIIH model, we carefully incorporated known structural or biochemical restraints. Specifically, the model incorporates all previously characterized CSB–CSA and CSA–UVSSA domain interactions (i.e., CIR and CIM motifs) that were identified either biochemically or observed in cryo-EM[33,47,48]. Protein interfaces were further refined using a combination of Rosetta docking and molecular dynamics simulations. In aggregate, our model elucidates not only the structural basis for cooperative CSB–CSA–UVSSA association but also how the complex ensures productive TFIIH association in a conformation capable of DNA unwinding.

In the TCR complex, TFIIH engages the forward-facing side of Pol II and is positioned downstream of the transcription bubble (Fig. 1a, Supplementary Movie 1). Notably, the overall conformation and binding mode of TFIIH closely resemble the PIC (Supplementary Fig. 2). In both assemblies, TFIIH's XPB translocase subunit binds the downstream DNA duplex and is poised to unwind and push the DNA toward the Pol II cleft. The analogy to PIC goes further when we consider the principal structural elements bridging the RNAPIIo and TFIIH. In the PIC, the crux of the Pol II–TFIIH interface is formed by the general transcription factor TFIIE, which is composed of two modules – the compact structured TFIIEβ, and the extended flexible TFIIEα. While TFIIEβ binds primarily to core PIC, the long flexible TFIIEα wraps around TFIIH and is principally responsible for TFIIH recruitment

(Supplementary Fig. 2a). Specifically, an acidic patch from TFIIEα forms an anti-parallel β-sheet with the PH domain of the p62 subunit. In the TCR–TFIIH complex, a tripartite module comprised of CSB, CSA, and UVSSA's VHS domain wedges itself between Pol II's Rpb1 subunit and TFIIH's XPB subunit, forming a critical bridge between RNAPIIo and TFIIH (Fig. 1b, Supplementary Fig. 2, Supplementary Movie 1). In this respect, the module could be considered a functional counterpart of TFIIEβ from the PIC. Elongation factor ELOF1 additionally stabilizes the TCR complex by anchoring CSA and UVSSA to Pol II without making contact to the incoming TFIIH (Fig. 1a, Supplementary Movie 1). ELOF1 binding also orients CRL4[CSA] for productive Pol II ubiquitination and indirectly prevents the DDB1 β-barrel assembly from clashing with the incoming TFIIH (Supplementary Fig. 3).

UVSSA features two principal domains, a VHS domain and a DUF domain (conserved domain of unknown function) interspersed by long flexible linkers (Supplementary Fig. 4). An intricate protein interactions network involving CSA, DDB1 and ELOF1 positions the VHS domain near the Pol II DNA entry tunnel (Fig. 1, Supplementary Fig 1a). While the VHS and portions of the DUF were recently visualized by cryo-EM[47,48], the TIR region remained unresolved. For this unresolved DUF segment, AlphaFold2[63,64] predicts a well-defined helical bundle (Supplementary Fig. 4) flanked by linkers – one to the VHS domain (residues 142-153); the other to the PHD interaction motif (residues 408-412). Considering the linker lengths, the DUF helical bundle is oriented towards the downstream DNA duplex and TFIIH's XPB subunit. Notably, with its modular architecture and flexible linkers UVSSA could be considered as a functional mimic of TFIIEα, which engages

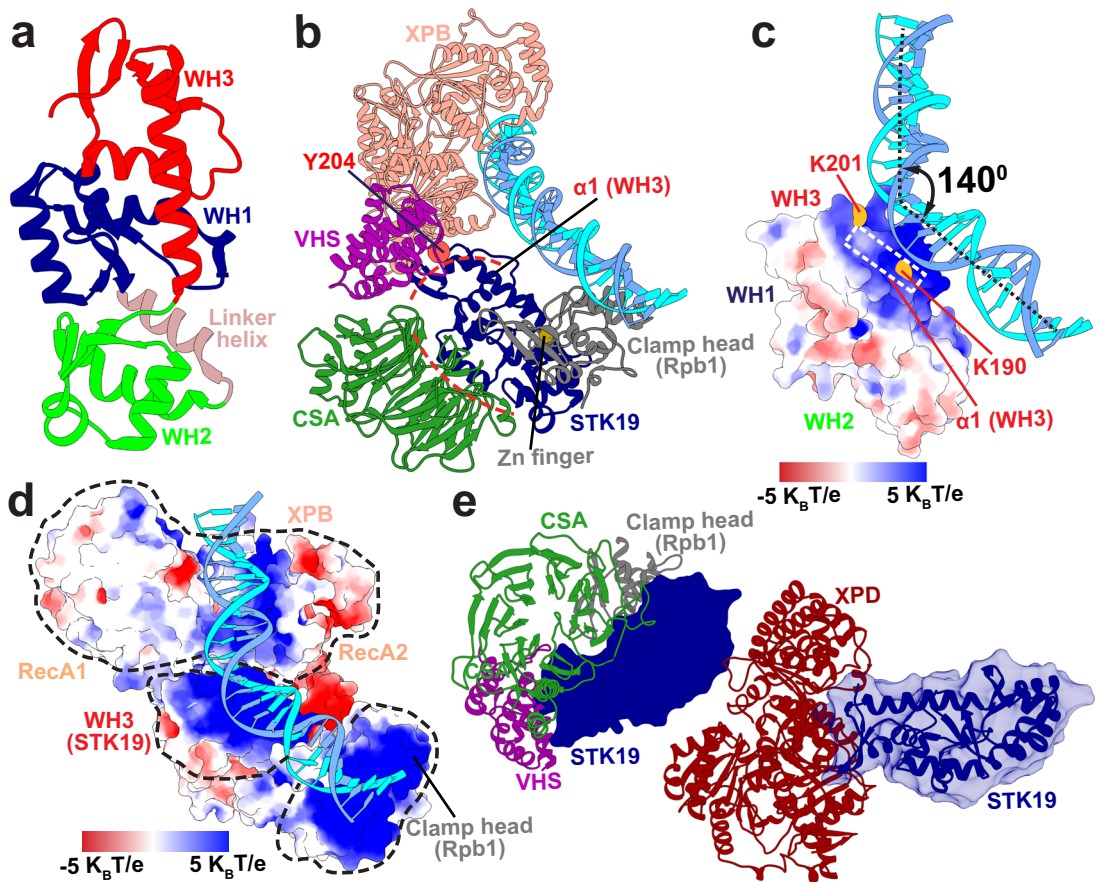

**Fig. 2 | STK19 is the linchpin of the TCR–TFIIH assembly and bridges RNAPIIo with core TCR factors and dsDNA. a** The structure of STK19 comprised of the three winged helix domains, WH1 in dark blue, WH2 in green, and WH3 in red. The linker helix connecting WH1 to WH2 is shown in light brown. **b** AlphaFold3 multimer model encompassing CSA in dark green, the VHS domain of UVSSA in purple, STK19 in dark blue, the clamp head of Rpb1 in gray, and XPB in pink. STK19–CSA and STK19–VHS interfaces are outlined by red dashed lines. The DNA-interacting α1 helix from WH3 and the zinc finger at the STK19–Rpb1 interface are labeled. **c** Electrostatic potential mapped onto the STK19 molecular surface. A scale bar of the electrostatic potential from negative (red) to positive (blue) is shown as an inset. The WH3 α1 helix is outlined by a white rectangle. Conserved residues forming salt bridge interactions with dsDNA are shown as yellow spheres and labeled. The kink in dsDNA induced by STK19 is shown by a black double-headed arrow. The bending angle is labeled. **d** A view of the electropositive groove formed by XPB, STK19, and the clamp head of Rpb1 that accommodates the downstream DNA duplex. XPB, STK19, and Rpb1 are outlined by black dashed lines. A scale bar of the electrostatic potential from negative (red) to positive (blue) is shown as an inset. **e** The STK19–XPD interface predicted by AlphaFold2 is shown. For comparison, the position of STK19 in the TCR–TFIIH complex is indicated by opaque blue surface relative to CSA (green), VHS (purple), the Rpb1 clamp head (gray), and XPD (dark red).

TFIIH via the p62 subunit (Fig. 1b, Supplementary Fig. 2). Like TFIIEα, UVSSA's TIR features a conserved acidic patch (residues 408-412) that forms a β-sheet with p62's PHD[56]. Thus, flexible TFIIH tethering to the core of the DNA processing machinery via the PHD is a defining feature of both transcription initiation and transcription-coupled repair.

A second key segment of the interface is provided by TFIIH's MAT1 subunit, which is positioned between the Pol II stalk region (Rpb4/7) and XPD (Fig. 1c, Supplementary Movie 1). In this orientation, MAT1's RING domain contacts Rpb4 and Rpb7 while MAT1's Arch anchor domain bridges the XPD Arch domain and Rpb4. While this binding mode is similar to the PIC, the MAT1 RING domain also engages one of the winged helix domains (WH2) of STK19. As part of the CAK module, MAT1's presence within the assembly is inhibitory to TCR and requires the subunit's displacement prior to lesion scanning and incision. Yet, in early TCR, MAT1 provides stabilization and orients the incoming TFIIH assembly. Thus, TFIIH is first recruited in an open repair-inhibited conformation, which requires subsequent CAK module removal and conformational closure to process damaged DNA.

### STK19 is the linchpin of the TCR-TFIIH complex

The most prominent features of the RNAPIIo–TFIIH interface are the CSB–CSA–VHS module and the XPD–MAT1–Rpb4/7 juncture.

Intriguingly, between these two structural elements, we observe a cavity well-suited to accommodate the STK19 protein, which encompasses three tightly packed winged-helix domains (WHs) (Fig. 2a) each featuring three α helices (α1, α2, and α3) and three β strands (β1, β2, and β3). The three WH domains are approximately linearly aligned, with WH1 occupying the central position. To situate STK19 within the recognized cavity, we first evaluated the propensity of STK19 to form binary interfaces with potential protein partners flanking the central cavity using AlphaFold2-multimer calculations[63,64]. We identified the Pol II clamp head (from the Pol II Rpb1 subunit), CSA, and the VHS domain of UVSSA as potential binding partners and, again using AlphaFold2, verified that all three proteins could bind STK19 concurrently. Positioning STK19 into the TCR–TFIIH complex based on the computed Rpb1, CSA, and VHS interfaces resulted in no clashes with either RNAPIIo or TFIIH. Strikingly, in this orientation, STK19 is poised to bind the downstream DNA duplex via a conserved electropositive patch of its WH3 domain. The significance of this interaction is highlighted by the fact that STK19 is a recognized sequence-independent double-strand DNA/RNA binding protein[57]. Yet, AlphaFold2 is unable to account for protein–DNA binding. To make further progress, we used AlphaFold3[65] to determine a larger complex composed of STK19, a segment of Rpb1, VHS, CSA, core TFIIH and dsDNA (Fig. 2b). The

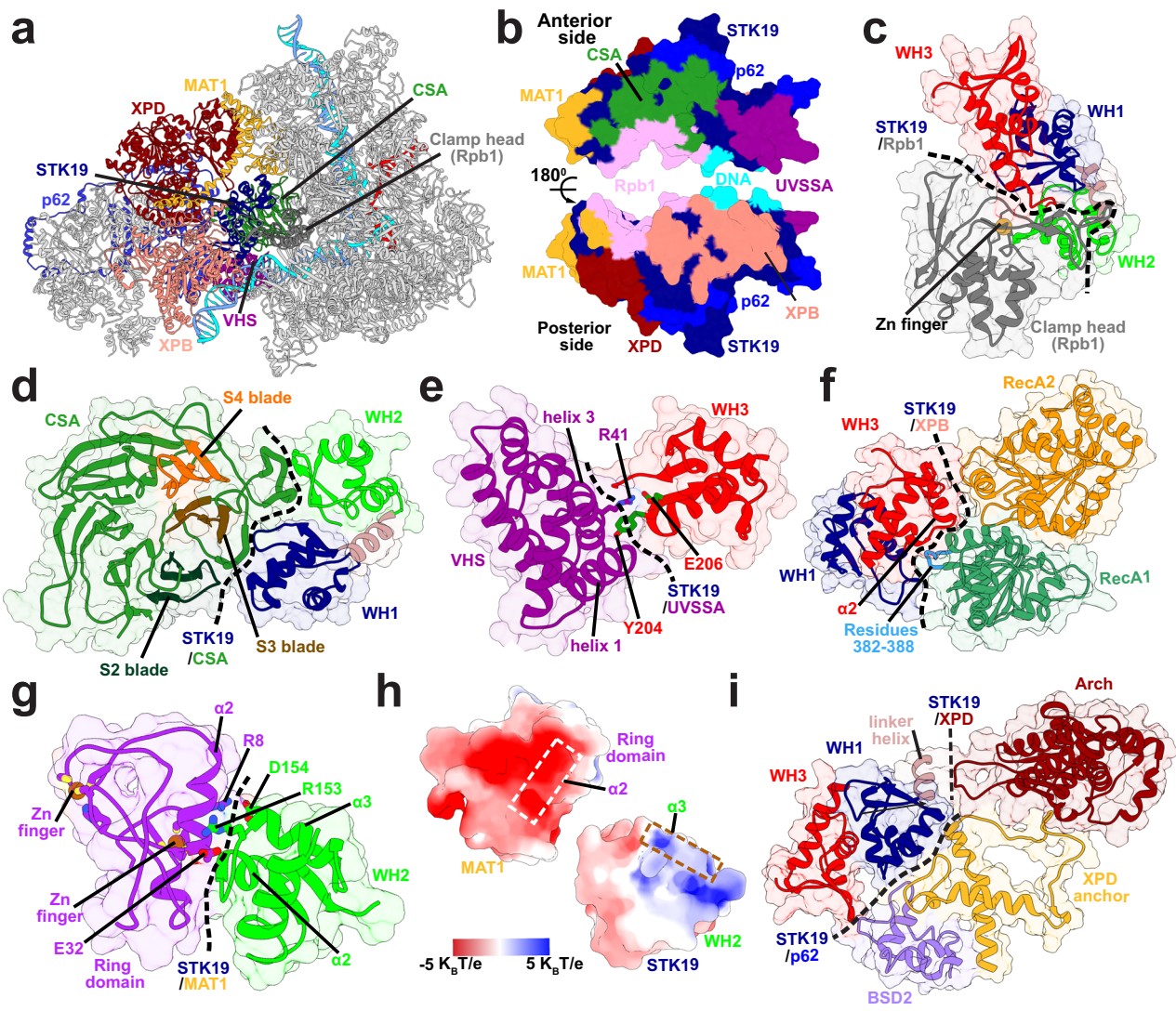

**Fig. 3 | STK19 serves as a landing platform for the incoming TFIIH by forming interfaces with XPB, XPD, p62, and MAT1. a** A view of the TCR−TFIIH complex highlighting the centrally positioned STK19. Proteins bound to STK19 are colored as follows: CSA (dark green), UVSSA (purple), MAT1 (gold), XPD (dark red), XPB (pink), p62 (medium blue), and Rpb1 (gray). **b** Anterior and posterior views of the STK19 molecular surface with interfaces to protein partners color coded and labeled. Zoomed-in views of STK19 interacting with core TCR proteins and subunits of TFIIH are shown for: **c** Rpb1 clamp head of Pol II; **d** CSA; **e** the UVSSA VHS domain; **f** XPB; and **g** the RING domain of MAT1. Key structural elements and persistent contacts at these interfaces are labeled and color-coded. **h** Surface electrostatic potential mapped onto the STK19-MAT1 interface. Dashed rectangles indicate key α-helices forming the interface. **i** STK19 Interfaces with the p62's BSD2 and XPD anchor domains.

complex showed STK19 positioned between the Pol II clamp head and RecA1 of XPB, forming together one continuous electropositive surface to accommodate dsDNA (Fig. 2c, d). We also note that AlphaFold2[61,62] (but not AlphaFold3) predicts STK19 binding to the back side of XPD near the interface of the Arch and Fe-S domains (Fig. 2e). However, the binding surface involved is the same that is responsible for VHS−STK19 binding, making STK19 association to XPD and UVSSA mutually exclusive. Moreover, STK19 placement based on XPD alone is incompatible with the rest of the TCR−TFIIH assembly. Thus, we find that multivalent interactions of STK19 with VHS, CSA, the Pol II clamp, XPB, and DNA collectively outweigh the STK19−XPD interaction.

Having positioned STK19 into the TCR−TFIIH model based on the AlphaFold3 results, we further optimized all protein interfaces using extensive 1-μs molecular dynamics simulations. The central position of STK19 in the complex (Fig. 3a, Supplementary Movie 1) highlights its

role as the linchpin of the TCR molecular machinery. STK19 forms interface with Rpb1, UVSSA, CSA, DNA and the TFIIH subunits XPB, XPD, p62, and MAT1 that completely encircle STK19 and, remarkably, cover >80% of its molecular surface area (Fig. 3b). Consistent with our integrative model and the idea of multivalent STK19 interactions, chromatin immunoprecipitation (ChIP) assays using STK19-GFP have shown STK19 engages multiple TFIIH subunits, not only XPD but also XPB and p62[60].

In our model, the anterior side of STK19 (residues 94-95, 119, 162-164 and 180-182) is anchored to RNAPIIo via the Pol II clamp head, which inserts a zinc finger into a conserved hydrophobic patch (residues 176-182) at the junction of STK19's three WH domains (Figs. 2b and 3c, Supplementary Movie 1). Two loops extending from the blades of the CSA β-propeller also project into the cleft between WH1 and WH2, establishing significant interactions with the α3 helix of WH1 and the β1 sheet of WH2 within STK19 (Fig. 3d, Supplementary Movie 1).

Moreover, UVSSA's VHS domain binds the β1 sheet of the WH3 domain near the STK19–DNA binding surface (Figs. 2b and 3e, Supplementary Movie 1). Two conserved interactions from this interface are notable: 1) the STK19 Y204 residue stacks between helix 1 and helix 3 of the VHS domain; and 2) additional interface stabilization arises from STK19's E206 residue forming a salt bridge with R41 from the VHS domain (Fig. 3e). While our model does not have a discernible STK19–CSB interface, the cooperative nature of the CSB–CSA–VHS association leads to enhanced anchoring of STK19 to the core of Pol II. The observed multivalent STK19 interactions explain why STK19 can pull down purified Pol II TCR complex lacking TFIIH in vitro[60].

**STK19 orients and stabilizes the incoming TFIIH**
Anchoring of the anterior side of STK19 to RNAPIIo allows the exposed posterior side to serve as a landing platform for the incoming TFIIH. XPB forms the largest interaction interface with STK19, accounting for ~50% of the total STK19–TFIIH buried surface area. XPB features two RecA-like helicase domains whose mutual displacement during ATP hydrolysis powers the translocation and remodeling of DNA in transcription or TCR. STK19 contacts primarily the RecA1 domain, which wedges itself against the WH1–WH3 domain junction (Fig. 3f, Supplementary Movie 1). WH3's α2 helix is key for establishing this interaction. Additionally, a RecA1 surface loop (residues 382-388) penetrates into a hydrophobic pocket located between STK19's WH1 and WH3 domains. In this orientation, STK19 associates with both XPB and the duplex DNA extending from the XPB DNA-binding cleft. Detailed mapping of the interactions between STK19 and XPB is shown in Supplementary Fig. 5a, b. Correspondingly, STK19 inserts directly onto the dsDNA path between XPB and the RNAPII clamp head. The DNA binding mode is via the minor groove and involves insertion of the α1 helix from the STK19 WH3 domain (Fig. 2b, c). Formation of this interface is guided by favorable electrostatics (Fig. 2c) and is mediated by salt bridge interactions involving conserved WH3 residues. Most prominently, the K190 and K201 residues bind across the minor groove precisely matching its width (Fig. 2c). Mutations of these residues are known to impair STK19's DNA binding[57]. In our optimized TCR–TFIIH model, the predicted STK19 association induces a ~140° kink in the DNA duplex between the Pol II clamp head and XPB (Fig. 2c and d), suggesting a role for STK19 not only in stabilizing the TCR–TFIIH complex but also in facilitating XPB unwinding by guiding dsDNA toward the Pol II cleft.

Surprisingly, TFIIH's MAT1 subunit binds the WH2 domain of STK19 opposite from the VHS binding site (Fig. 3g, Supplementary Movie 1). We note that MAT1's RING domain shifts significantly during the MD simulations, establishing persistent contacts to WH2 primarily through salt-bridge interactions (e.g., R153-E32 and D154-R8 residue pairs). The interface resembles a helical bundle involving the α2 helix of the MAT1 RING domain and the α2, α3 helices of WH2 of STK19. Moreover, the interaction surface exhibits high electrostatic complementarity (Fig. 3h). Consequently, loss of contacts upon MAT1 removal during the latter stages of TCR may destabilize STK19 and cause its ultimate dissociation or relocation within the TCR assembly.

The p62 subunit of TFIIH interacts with STK19 through the BSD2 and XPD anchor domains (Fig. 3i, Supplementary Movie 1). Specifically, BSD2 and the connecting linker to BSD1 engage the WH1 and WH3 domains of STK19. The helices of the XPD-anchor domain of p62 contact primarily WH1, establishing a key connection between STK19 and XPD. This observation carries functional implications. In the PIC, a loop from the XPD-anchor region of p62 inserts into the DNA binding groove of XPD and inhibits its ssDNA translocase activity. By contrast, XPD's activity is required for TCR. The fact that STK19 associates with the inhibitory XPD-anchor loop implies a potential regulatory role for STK19 whose displacement or relocation during the latter stages of TCR may cause the removal of the inhibitory lock on XPD. An anchoring loop from XPD's Arch domain (residues 272-299) is the only

element of XPD that directly associates with STK19 via its linker helix. This interaction may also be involved in p62 detachment and XPD translocase activation in the latter stages of TCR. A detailed view of the interaction interface between STK19 and p62, highlighting the specific residues involved, is provided in Supplementary Fig. 5c and d.

**Functional roles of CRL4$^{CSA}$, ELOF1, and RNAPIIo ubiquitination in TCR–TFIIH recruitment**
Recruitment of TFIIH to the evolving TCR machinery is critically dependent on the ubiquitination of RNAPIIo and UVSSA. In response to UV irradiation, the Pol II Rpb1 subunit is polyubiquitinated at residue K1268 by the CRL4$^{CSA}$ complex. In turn, this process triggers mono-ubiquitination of UVSSA and stimulates TFIIH binding. To achieve specific ubiquitination, CRL4$^{CSA}$ must be positioned precisely to interact with Rpb1 within the nascent TCR assembly. To uncover the structural basis for this precise positioning, we modeled the TCR complex bound to CRL4$^{CSA}$ based on the cryo-EM structures of core TCR (PDB ID: 8B3D) and CRL4$^{CSA}$ (PDB ID: 8B3I). We then performed microsecond-timescale MD simulations of the TCR–CRL4$^{CSA}$ complex and several subcomplexes: TCR–DDB1, TCR–DDB1–VHS and TCR–DDB1–VHS–ELOF1. We compare representative MD conformers (Fig. 4a–c) and the relative flexibilities of these complexes (Fig. 4d–f) by mapping computed B-factors from MD onto the structural models. We also evaluate the key conformational shifts that occur during the stepwise addition of VHS and ELOF1 to the growing TCR–CRL4$^{CSA}$ assembly (Fig. 4g and h). We first note the elevated mobility of CSA and DDB1 in the absence of UVSSA or ELOF1. CSA serves as the substrate-recognition subunit of CRL4$^{CSA}$, which also includes the three β-propeller domains of DDB1 (denoted BPA, BPB, and BPC). The modest CSA–CSB interface established through the CIM motif permits wide swing motions of DDB1 with respect to the TCR core leading to high B-factor values for this region. The mobility of CSA and DDB1 is markedly reduced upon VHS binding due to anchoring interactions that the VHS domain establishes with CSA and the downstream DNA duplex. Furthermore, addition of ELOF1 to the assembly completely suppresses the residual flexibility of DDB1–CSA, yielding exceptionally low B-factors (Fig. 4f). The pronounced reduction in CSA mobility in the presence of ELOF1 was also independently observed by cryo-EM[47]. We observe a ridge of stability that extends from RNAPIIo and encompasses CSB, CSA, two of the DDB1 β-propeller domains (BPA and BPC), and the VHS domain of UVSSA. Notably, the BPB β-propeller remains mobile, which is key for enabling the swinging motion of the Cullin4A arm of the CRL4$^{CSA}$ ubiquitin ligase complex. Importantly, ELOF1 was key for the observed stabilization of TCR–CRL4$^{CSA}$ due to formation of an extended ELOF1-CSA binding surface centered on the ELOF1 zinc finger and positioned opposite to DDB1 on CSA. Moreover, ELOF1 strongly interacts with the UVSSA VHS domain, enhancing the existing UVSSA association through the CIR motif. Given the crucial role of UVSSA for the recruitment of TFIIH and all downstream TCR factors, the stabilizing effect of ELOF1 on the assembly explains why endogenous UVSSA association is greatly reduced in ELOF1-KO cells[52]. The combined ELOF1–CSA–VHS interface (Fig. 4g) causes a ~7 Å shift of the VHS domain along the downstream DNA duplex relative to the TCR–DDB1 complex (Fig. 4h). This is accompanied by ~35° rotation of the entire VHS–CSA–DDB1 module around the dsDNA axis (Fig. 4h). By bridging Rpb1, CSA and UVSSA together, ELOF1 orients and firmly anchors CRL4$^{CSA}$ to RNAPIIo. Thus, our results highlight the critical roles of CSA, the UVSSA VHS domain, and ELOF1 in establishing a ubiquitin ligation-competent TCR complex.

**ELOF1 arrests the Cullin4A arm rotation to promote specific Rpb1-K1268 ubiquitin transfer**
Besides CSA and DDB1, CRL4$^{CSA}$ E3 ubiquitin ligase encompasses Cullin4A and Ring Box Protein-1 (RBX1). The Cullin4A protein consists primarily of hydrophobic heat repeats whose flexibility enables close

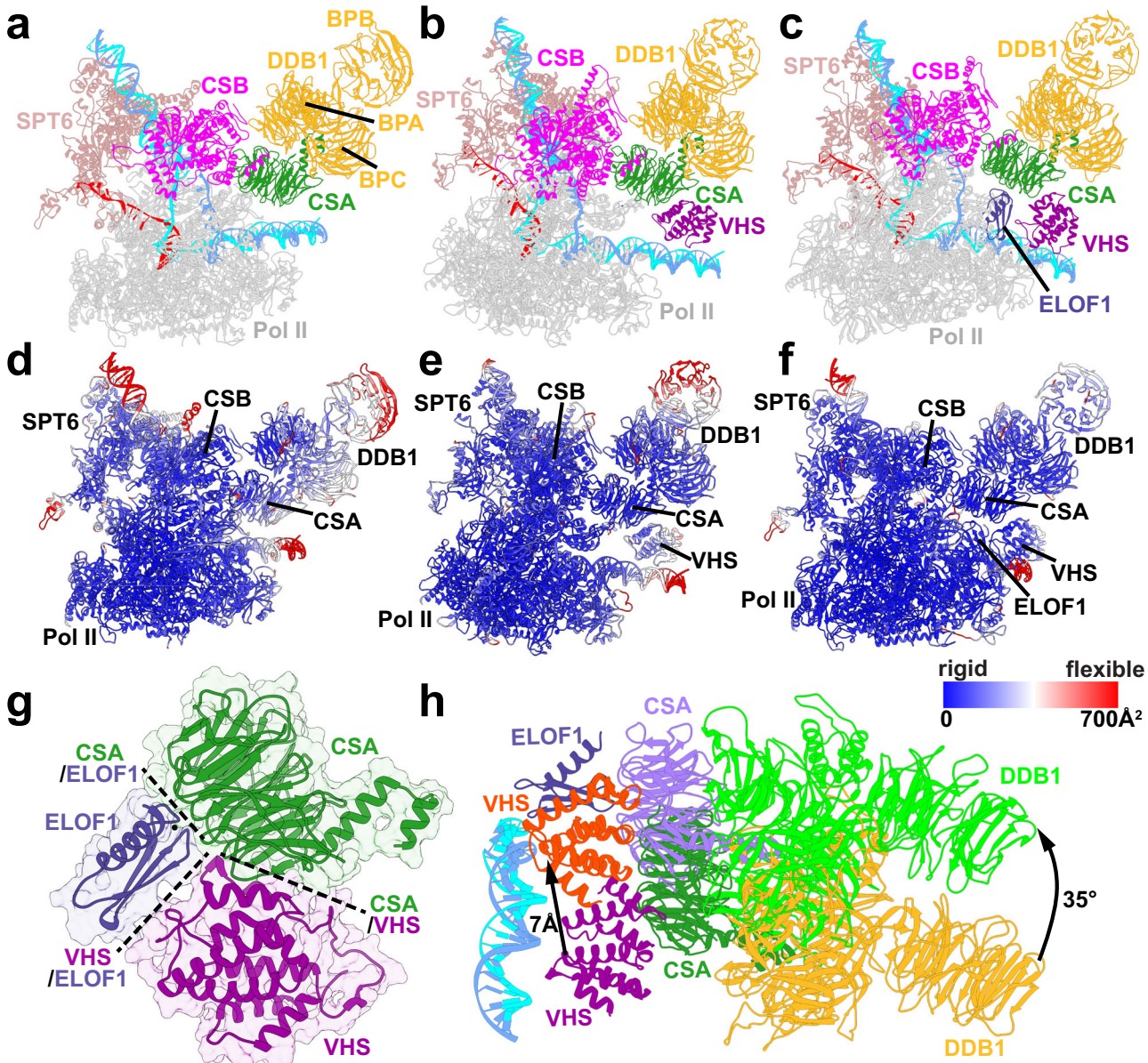

**Fig. 4 | ELOF1 suppresses CSA mobility and repositions UVSSA's VHS domain to facilitate precise CRL4^CSA assembly.** Integrative structural models of (**a**), the TCR–CSA–DDB1, (**b**), TCR–CSA–DDB1–VHS, and (**c**), TCR–CSA–DDB1–VHS–ELOF1 complexes. Proteins are labeled and color-coded as follows: CSB (magenta), CSA (green), DDB1 (gold), UVSSA (purple), SPT6 (light brown), Pol II (gray). Template and non-template DNA is shown in cyan and blue; RNA is in red. Computed B-factors are mapped onto the structures of (**d**), the TCR–CSA–DDB1, (**e**), TCR–CSA–DDB1–VHS, and (**f**), TCR–CSA–DDB1–VHS–ELOF1 complexes. Rigid and flexible regions are color-coded from blue to red, respectively. **g** Zoomed-in view of the CSA–VHS–ELOF1 module. Protein interfaces are indicated by black dashed lines. **h** Comparison of the positioning of the CSA–VHS–DDB1 module in the presence/absence of ELOF1. In the model without ELOF1, CSA, VHS, and DDB1 are colored in green, gold, and purple, respectively. In the model with ELOF1 (dark purple), CSA, VHS, and DDB1 are colored in violet, orange, and light green. The extent of the conformational transition between the two states is indicated by black arrows.

positioning of the Ub-targeted lysine to the RING-bound E2 ubiquitin-conjugating enzyme, which then transfers ubiquitin onto the substrate. Once assembled on RNAPIIo, the CRL4^CSA complex undergoes a dramatic conformational shift, which enables the Cullin4A arm to reach the Rpb1 K1268 position. To reveal the impacts of Cullin4A conformational switching on the functional dynamics of the TCR assembly, we performed microsecond-timescale molecular dynamics simulations. We first built and simulated models of TCR–CRL4^CSA in three distinct conformations of the BPB domain of DDB1: linear, hinged, and twisted (Fig. 5a). Identified in a recent cryo-EM study[74], the three conformers reflect the discrete flexibility of the BPB β-propeller. The high mobility of BPB in our simulations could be attributed to the small number of persistent contacts between that domain and the

other two DDB1 β-propellers. The combination of discrete BPB flexibility and the continuous flexibility of the Cullin arm (Fig. 5b) allowed the E3 ligase to sample three distinct regions on the surface of the core-TCR assembly proximal to 1) the UVSSA VHS domain (Fig. 5c), 2) the Rpb1 and Rpb9 subunits of RNAPIIo (Fig. 5d) and 3) CSB (Fig. 5e). In the linear conformer, the Cullin arm was highly mobile and had minimal interactions with the VHS (Fig. 5c). By contrast, in the hinged and twisted orientations, the Cullin arm established interfaces with RNAPIIo (Fig. 5d) and CSB (Fig. 5e) that were stable over the microsecond duration of the simulations. After thoroughly equilibrating the linear, hinged, and twisted conformations in unbiased MD runs, we used the partial nudged elastic band (PNEB) method[75] to simulate the full rotation of the Cullin arm in the TCR–CRL4^CSA complex. PNEB yields

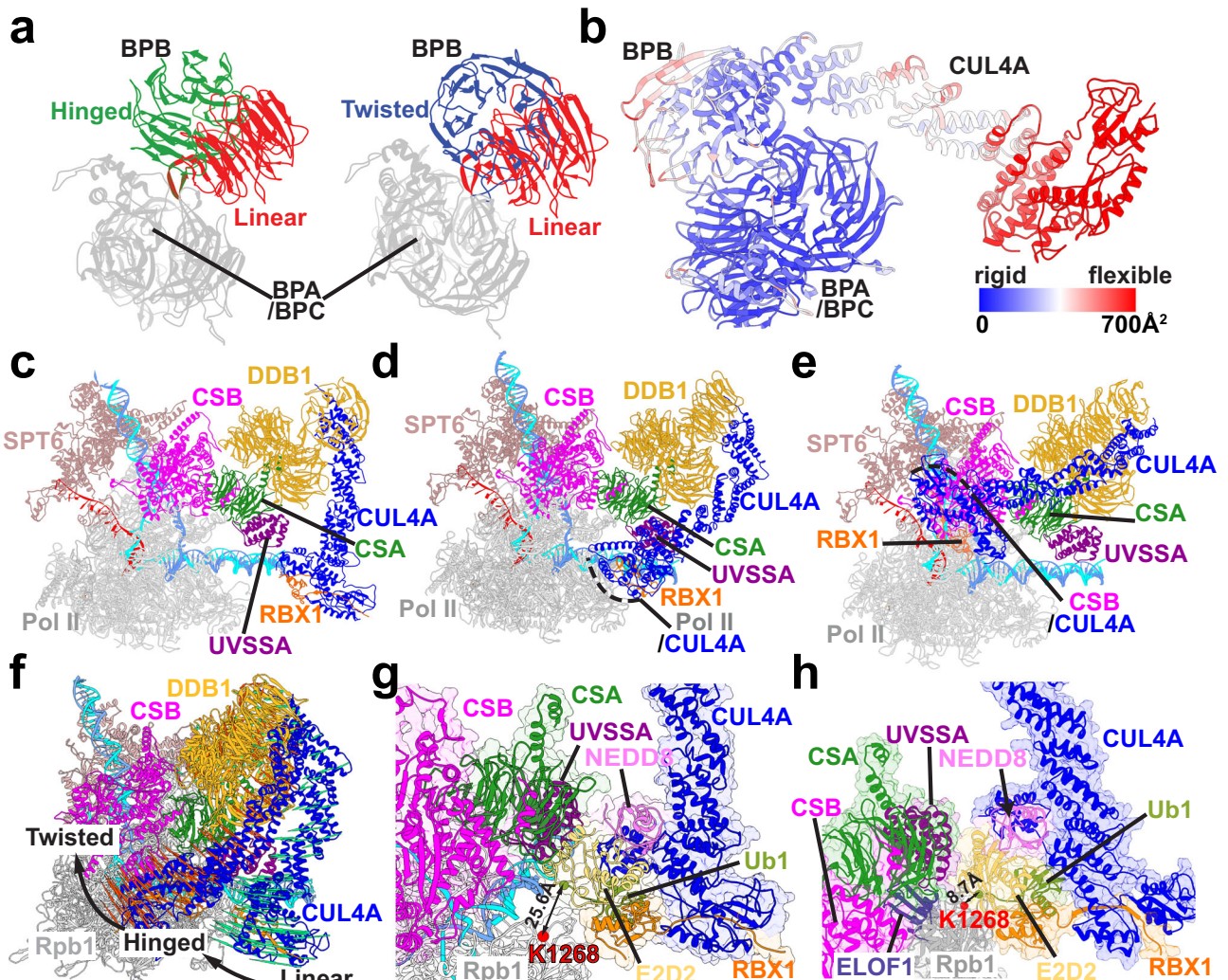

**Fig. 5 | ELOF1 arrests the Cullin4A arm rotation and positions CRL4^CSA for specific ubiquitin transfer. a** The BPB β-propeller domain of DDB1 shown in the hinged (green), twisted (blue), and linear (red) conformations. The BPA and BPC β-propeller domains of DDB1 are shown in gray. **b** B-factor values mapped onto the CRL4^CSA structure and color-coded from blue to red to indicate rigid versus flexible regions, respectively. **c–e** Views of the CRL4^CSA complex in (**c**), the linear, (**d**), hinged, and (**e**), twisted BPB domain conformations. Interfaces formed with CSB and Rpb1 are indicated by black dashed lines. **f** A porcupine plot showing the highly directional movement of the Cullin4A arm that enables Rpb1 and CSB ubiquitination. The conformational transition follows the optimal MFEP path from PNEB. **g, h** Magnified views of the E2D2, NEDD8 and Ub1 near the K1268 ubiquitination site of Rpb1: (**g**), in the absence of ELOF1 and (**h**), in the presence of ELOF1. The Cα-Cα distances between K1268 of Rpb1 and G76 of Ub1 are shown.

minimal free energy paths (MFEP) for this conformational transition represented as a series of replicas of the simulation system. The replicas encapsulate the entire conformational switching mechanism, including all on-path intermediates.

Productive ubiquitination requires cooperative action by the DDB1–CUL4A–RBX1 E3 ubiquitin ligase and the E2-ubiquitin ligase, which includes the E2D2 ubiquitin-conjugating enzyme[47]. In particular, proximity of the targeted lysine residues to the C85 residue in the E2D2 active site is essential for the ubiquitin transfer reaction. In our modeling, in addition to E2D2 and ubiquitin (Ub1), we also incorporate NEDD8–a ubiquitin-like moiety essential for CRL4^CSA activation in vivo[47]. The PNEB results recapitulate the global motions of CRL4^CSA as it transitions from the linear to the hinged and twisted BPB conformations (Fig. 5f, Supplementary Movie 2). The conformational ensembles from the PNEB simulations reveal multiple potential ubiquitination sites on both Rpb1 and CSB that are accessible to the ubiquitin ligase and fall within an optimal distance range (<12 Å) from the E2D2 active site. Notably, seven lysine residues on CSB within reach of the Cullin arm were identified as potential ubiquitination sites (CSB residues

K725, K729, K747, K751, K759, K988, and K991), including two previously validated ubiquitination sites, K729[76] and K759[77]. This outcome poses a conundrum regarding the involvement of CRL4^CSA in TCR: precisely how does the E3 ligase complex target a single lysine residue, Rpb1 K1268, among other possible sites? Notably, in the absence of ELOF1, none of the intermediates observed along the PNEB pathway position the K1268 residue sufficiently close to the E2D2 C85 residue to enable efficient ubiquitination. The smallest observed Cα distance between the terminal glycine residue of the transferred ubiquitin and the Rpb1 K1268 residue was 25.6 Å (Fig. 5g). Moreover, the fraction of conformations of the Cullin arm approaching the K1268 position is minimal. When released in unbiased MD simulation runs, the Cullin arm quickly moves past the K1268 position to interact with CSB or other regions of RNAPIIo. Thus, in the absence of ELOF1 CRL4^CSA is not positioned to efficiently or specifically ubiquitinate Rpb1. By contrast, the presence of ELOF1 induces the formation of a specific complex in which the distance required for ubiquitin transfer is markedly reduced to 8.7 Å (Fig. 5h, Supplementary Movie 2). Our study corroborates previous findings[47] from cryo-EM that established the role of ELOF1 in

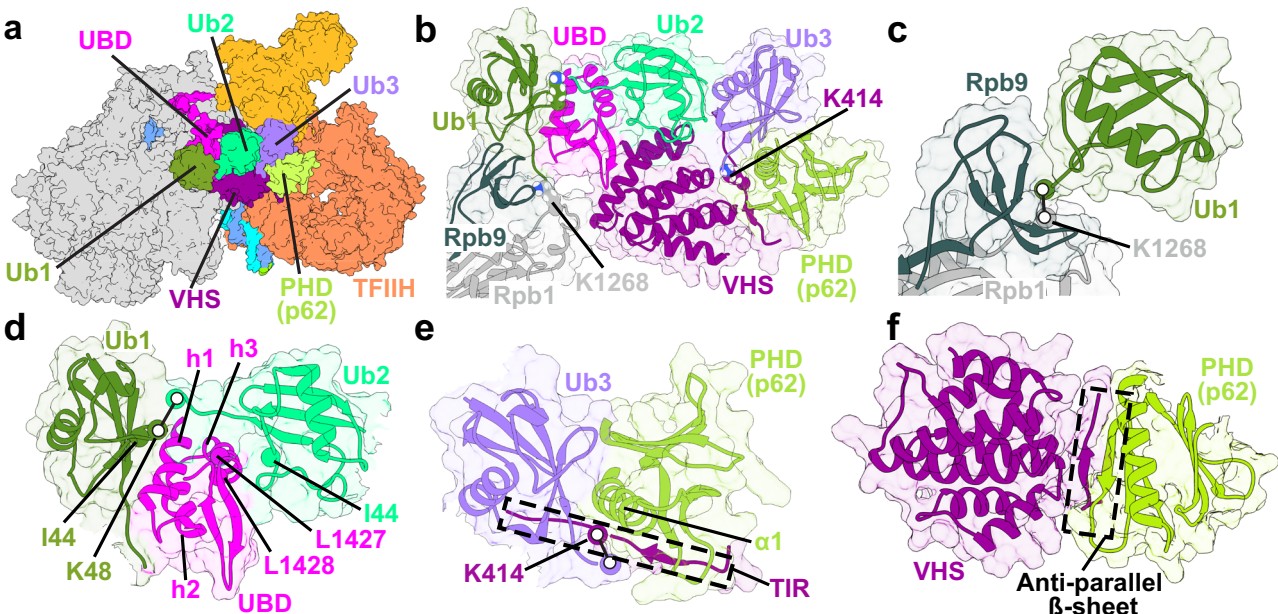

**Fig. 6 | K1268-polyubiquitination of Rpb1 and the UBD of CSB strengthen the contacts of VHS with the p62 PH domain and K414-ubiquitinated UVSSA to enable TFIIH recruitment. a** Integrative model of the TCR–TFIIH–Ub complex shown in surface representation. The three ubiquitin moieties in the model are colored as follows: Ub1 in olive, Ub2 in green, and Ub3 in medium purple. The UBD is shown in magenta, the UVSSA VHS domain in purple. The PH domain of p62 is in light green, and TFIIH is depicted in orange. **b** Zoomed-in view of Ub1, Ub2, and Ub3 interacting with Pol II Rpb1, Rpb9, the UBD of CSB, PHD, and UVSSA VHS domain. The sites of ubiquitin conjugation, Rpb1-K1268, and UVSSA-K414, are shown as van-der-Waals spheres. **c** Zoomed-in view of Ub1 conjugated to the K1268 site on Rpb1. **d** the CSB UBD interacting with the K48-linked Ub1 and Ub2. Key structural elements and residues at the interfaces are labeled and colored by domain. **e** Ub3 is conjugated to K414 on UVSSA and interacts with the p62 PH domain and UVSSA. **f** Interaction between the PHD, the TIR, and the VHS domain of UVSSA.

positioning CRL4$^{CSA}$ for specific ubiquitination. Indeed, the same study solved the separate cryo-EM structures of core TCR–ELOF1 (8B3D) and CRL4$^{CSA}$ (PDB ID: 8B3I) but did not put forward a composite model for TCR–CRL4$^{CSA}$. The reason is that when the 8B3D and 8B3I structures are superimposed on either CSA or DDB1, E2D2 clashes with the VHS domain, and RBX1 overlaps with the Rpb1 jaw. Thus, the precise distance from K1268 to the E2D2-bound Ub could not be ascertained. Here we extend these findings and show that the Cα distance reduction is not due to any direct contacts between ELOF1 and the E2-ubiquitin ligase. Instead, ELOF1 repositions the VHS domain of UVSSA and orients CRL4$^{CSA}$ by stacking VHS helix 6 against the C-terminal α4 helix of E2D2. Concurrently, the E2D2 α3 helix engages the surface loop of Rpb1 harboring the K1268 residue (Supplementary Fig. 6). Next, we simulated this arrested ubiquitination-competent state in free unbiased MD (Fig. 5h, Supplementary Movie 2). Notably in our simulated model, the DDB1 BPB β-propeller conformation was distinct from the previously recognized linear, hinged, or twisted orientations. Instead, the conformer closely resembled one of the transient intermediates observed in the PNEB pathway between the linear and hinged states. Importantly, removal of ELOF1 from the complex abolished the stability of this conformer, forcing BPB to quickly adopt either the linear or hinged conformations in multiple independent released MD trajectories. Thus, our results unveil the protein interactions and dynamics of core TCR factors (e.g., ELOF1) and the subunits of the DDB1-CUL4A-RBX1 ubiquitin ligase leading to productive Rpb1 ubiquitination.

### Ubiquitination of Rpb1 and UVSSA cooperatively facilitate TFIIH recruitment via p62's PHD

To address the question of how K1286-ubiquitinated Rpb1 and K414-ubiquitinated UVSSA facilitate TFIIH recruitment after the departure of the E2 ubiquitin ligase, we modeled the TCR–TFIIH complex carrying the respective ubiquitin modifications. First, we used the conjugated protein docking protocol in the Rosetta package to optimally position

the K1286-bound ubiquitin (Ub1) in our TCR–TFIIH model. Conjugated docking afforded optimal sampling of Ub1 interactions with all protein chains adjacent to the ubiquitination site (Rpb1, Rpb9, ELOF1, and VHS) while maintaining the covalent attachment of the Ub C-terminal glycine residue to K1286. We then clustered the ensemble of docking poses generated by Rosetta. From the cluster centroids, we selected the lowest energy docking pose, which was subsequently used to build the Ub–TCR–TFIIH model. Furthermore, we recognized that the C-terminal end of CSB (residues 1386-1491) harbored a ubiquitin binding domain (UBD)[54,55,78,79] that could stabilize a polyubiquitin chain starting at the K1268 position. Importantly, RNAPIIo ubiquitination and TFIIH recruitment are both impaired in CSB$^{ΔUBD}$ cells, showing that CSB UBD is key for functional TCR[55]. Conversely, replacement of UBD with the UBA domain of Rad23[54] (Supplementary Fig. 7) reactivates TCR. To model polyubiquitin chain association, we took advantage of an existing NMR structure of di-ubiquitin bound to the UBA domain of human Rad23A (PDB ID: 1ZO6)[80]. By replacing UBA with UBD and superimposing the UBD–di-ubiquitin module onto the ubiquitinated Rpb1 we were able to integrate this structural element into the complete Ub–TCR–TFIIH model. Finally, we used AlphaFold2-multimer to create a structural model of p62 PHD bound to 1) the UVSSA TIR (residues 405-415); 2) the VHS domain, and 3) ubiquitin (Ub3) covalently attached to K414 of UVSSA. The new structural module was integrated into the Ub–TCR–TFIIH model by superimposing onto the VHS domain.

Our combined Ub–TCR–TFIIH model is presented in Fig. 6. Remarkably, we find that UBD and the three ubiquitin moieties (Ub1, Ub2, Ub3) associate to form an uninterrupted chain of globular domains spanning the VHS surface from the Rpb1/Rpb9 interface to the PH domain of p62 (Fig. 6a and b). The UBD is flanked by Ub1 and Ub2 (Fig. 6d) and connected to the CSB core via a linker extending toward the CIM (Fig. 6a). The UBD comprises three helices (h1, h2, and h3) and a β-hairpin that form a hydrophobic core centered on a conserved dileucine motif (residues 1427, 1428). Binding to Ub1 is

mediated by hydrophobic contacts between the UBD h1 helix and the canonical I44 interaction surface of ubiquitin. Conversely, Ub2 binds the UBD h3 helix and dileucine motif on the opposite side of UBD. Ub1 and Ub2 are the first two monomers of a K48-conjugated ubiquitin chain that extends from the Rpb1 K1268 residue (Fig. 6c) and is stabilized by the C-terminal UBD of CSB and the VHS of UVSSA, which is itself a ubiquitin-binding domain. Notably, a previous experimental study showed the formation of both K48 and K63-linked poly-Ub chains at this site[50]. The polyubiquitin chain is poised to interact with a third ubiquitin (Ub3) conjugated to K414 of UVSSA. The UVSSA TIR harbors an acidic patch, which forms an anti-parallel β-sheet with the p62 PHD (Fig. 6e and f)[56]. This binding mode is analogous to XPC–PHD interaction in GG-NER[81], which is also mediated by a conserved acidic patch on XPC. Thus, the mechanism of initial TFIIH recruitment is partly shared by TCR and GG-NER. Moreover, based on AlphaFold2 modeling we show that Ub3 binds directly to the p62 PH domain (Fig. 6e). The central helix of the PHD inserts and forms contacts with the canonical hydrophobic patch of Ub3. AlphaFold2 also predicts stable complex formation between the PHD and the VHS domain of UVSSA (Fig. 6f). Collectively, our results shed light on the interplay of RNAPIIo- and UVSSA- ubiquitin modifications and their distinct functional roles in TFIIH recruitment and TCR. We show that successful TFIIH recruitment to lesion-stalled RNAPIIo requires more than just flexible tethering via the PHD to UVSSA. Instead, stable attachment and orientation of TFIIH relies on a multitude of interacting protein domains. Thus, we posit a mechanism wherein the polyubiquitin chain and the UBD buttress the interactions of VHS with the p62's PH domain and K414-ubiquitinated UVSSA, forming an essential bridge between RNAPIIo and TFIIH.

### Global motions and dynamic modules of the TCR molecular machinery

To uncover the functional dynamics of the Ub–TCR–TFIIH complex, we performed microsecond-timescale MD simulations. We then employed dynamic network analysis[82–84] to analyze the resulting extensive conformational ensemble. Using covariance data from the MD simulations, network analysis partitions the complex into communities that represent all dynamically independent structural modules of the system – a description that transcends traditional subdivision by protein domains. Network analysis identifies 36 dynamic communities in Ub–TCR–TFIIH, which are color-coded and mapped onto the structure (Fig. 7a and b, Supplementary Fig. 8, Supplementary Movie 3). The edge betweenness graph (Fig. 7b) encodes the magnitude of allosteric communication between pairs of communities, yielding a coarse-grain view of the dynamic connectivity throughout the complex. In this view, the structural elements maintaining the integrity of the PNAPIIo–TFIIH interfaces are represented by strongly connected components of the protein network graph. Among these elements, the two CSB domains (community V and community M), CSA and ELOF1 (community H) form an exceptionally well-connected cluster, reflecting the stable CSB–CSA and CSA–ELOF1 association in our integrative model. The CSB–CSA–ELOF1 community cluster (Fig. 7c) establishes additional robust connections to STK19 (community N), which in turn anchors the cluster to RNAPIIo via Rpb1 (community B). Additionally, RecA2 of CSB (community V) is strongly coupled to Rpb2, establishing a second anchoring point to RNAPIIo. Lastly, the VHS domain of UVSSA (community 14) connects to community H (CSA, ELOF1) establishing a basis for the cooperative assembly of CSB, CSA, and UVSSA on lesion-stalled RNA polymerase II. Our findings highlight the remarkable connectivity of community N within the TRC–TFIIH–Ub assembly. Community N encompasses not only STK19 but also includes segments from MAT1's RING domain, the Rpb1 clamp, CSA, XPB and XPD, reflective of its central positioning and pivotal role in the assembly of the TCR machinery (Fig. 7d). Community N forms key connections to communities I (XPD, p62), H(CSA,

ELOF1), B(Rpb1), 2 (MAT1, XPD Arch), 5 and 7 (XPB). Importantly, STK19 serves as the principal bridge between Rpb1 of RNAPIIo (community B) and TFIIH's XPD and p62 subunits (community I), establishing STK19 as the linchpin of the TCR–TFIIH complex. MAT1's Arch anchor domain and the XPD Arch domain segregate into a single dynamic module (community 2), which is well-connected to STK19 (community N) and XPD/p62 (community I) as well as Rpb4/7 (community T). This observation highlights MAT1's importance serving as a bridge between TFIIH and RNAPIIo (Fig. 7d) whose disruption during the later stages of TCR could precipitate the repositing of STK19, p62's BSD2 domain and the Arch anchor helices and, thus, potentially activate XPD for lesion scanning. Intriguingly, our findings also shed light on the significance of Rpb1 and UVSSA ubiquitination for the integrity of the TCR complex (Fig. 7e). Ub1, Ub2, and the UBD domain of CSB come together to form a single dynamic module (community 3), which is the core of a closely interwoven community cluster that includes the VHS domain (community 14), CSA and ELOF1 (community H), BPC of DDB1 (community G), p62 PHD, TIR of UVSAA and Ub3 (community Z) and segments of Rpb1, Rpb9 and UVSSA (community U). This community arrangement reflects the role of Ub1, Ub2, and UBD in strengthening the VHS binding to Pol II, CSA, and ELOF1. In turn, VHS serves as a landing platform for the p62 PHD-Ub3-TIR module (community Z), which is responsible for the initial TFIIH tethering to UVSSA and for orienting TFIIH in the nascent TCR complex.

## Discussion

Transcription-coupled repair complexes are amazingly dynamic protein machines that actively reshape themselves and self-regulate to attain precise lesion removal from the transcribed genome. By combining existing structural data with advanced computational modeling, we built a practically complete integrative model of the TCR–TFIIH complex and analyzed its functional dynamics on timescales accessible to MD simulations. We also modeled and simulated TCR–CRL4$^{CSA}$ complexes in diverse functional states to elucidate the mechanisms of RNAPIIo and UVSSA ubiquitination. Our results shed light on the precise functional roles of numerous factors (notably, ELOF1, STK19, UVSSA, and polyubiquitin) in ensuring productive TFIIH engagement to the nascent TCR machinery.

Our computationally informed mechanism (Fig. 8), sheds light on the early-stage reorganization of the TCR protein machinery. Transcription-blocking lesions in the template strand are first recognized by Pol II which stalls at the lesion site. Thus, damage recognition in TCR is carried out by the RNA polymerase itself and does not require any specific damage recognition factors. Stalling of Pol II is a signal for the sequential cooperative recruitment of core TCR factors – CSB, CSA, and UVSSA to the nascent repair complex. CSB arrives first to the damage site and binds Pol II upstream of the transcription bubble[22]. CSB then serves as a molecular motor to power Pol II bypass of less bulky lesions[8,22]. Failure to bypass the lesion leads to CSB-mediated recruitment of CSA via its CIM motif[33], which is further stabilized by elongation factor ELOF1[52]. In turn, CSA cooperatively associates with UVSSA, which is ultimately responsible for bringing TFIIH to the growing repair complex[33]. TFIIH engages stalled RNAPIIo downstream of damage site and acts as a molecular motor to dislodge the lesion from the polymerase active site and allow repair factors to access it. CSA also functions as a substrate recognition subunit of CRL4$^{CSA}$, an E3 ubiquitin ligase whose activity introduces ubiquitin modifications onto Pol II and UVSSA[49,60]. Rpb1-K1268 and UVSSA-K414 ubiquitination are independently required for TFIIH recruitment to lesion-arrested RNAPIIo[50,58]. Our simulations of CRL4$^{CSA}$ uncover the interplay of segmental DDB1 flexibility and continuous Cullin4A flexibility in exposing wide swaths of the Rpb1 and CSB molecular surfaces to ubiquitin modifications. Previous biochemical and cryo-EM studies[47], had uncovered the role of ELOF1 in suppressing CSA mobility to facilitate

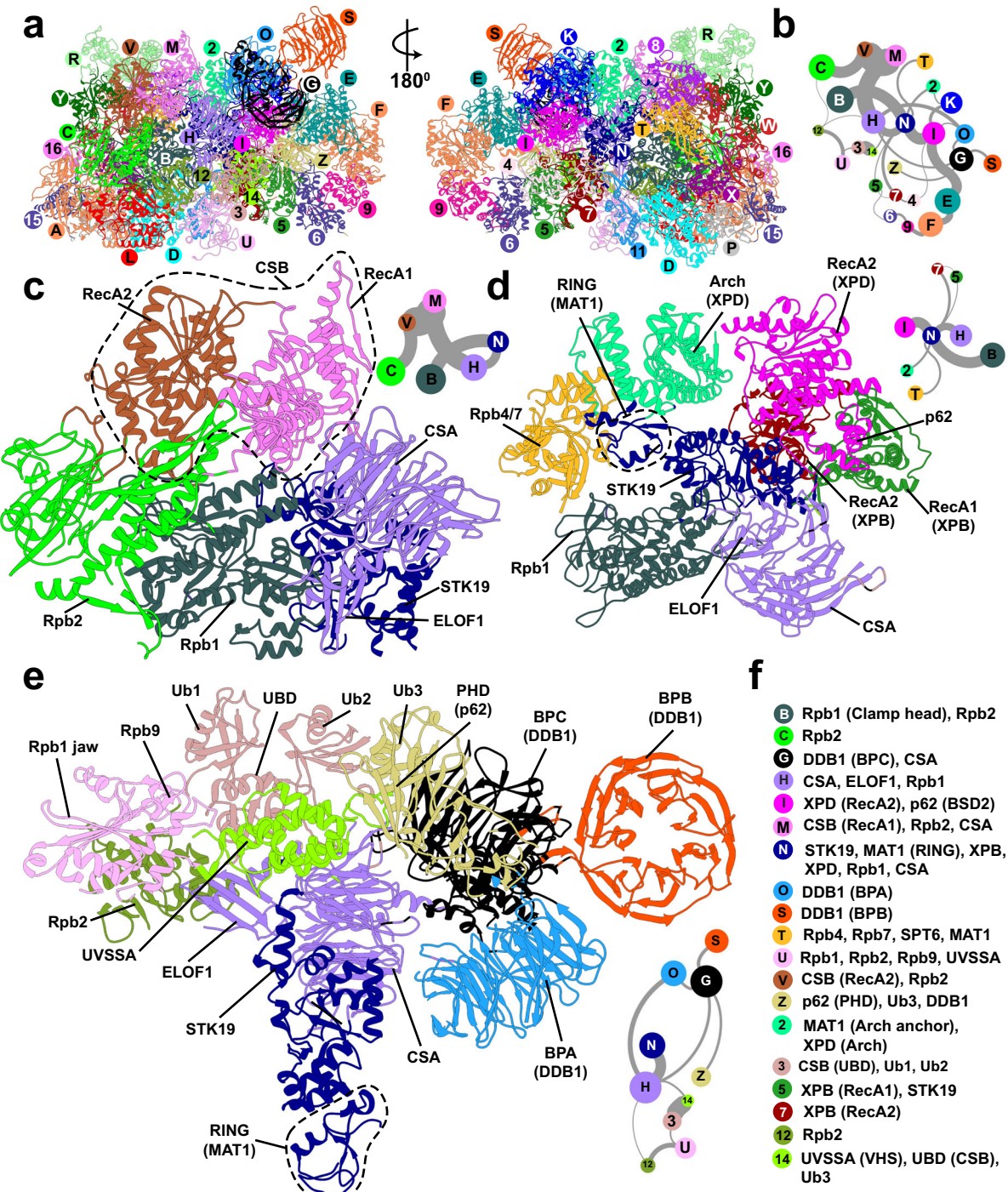

**Fig. 7 | Network of dynamic communities underlie TCR–TFIIH–Ub assembly's functional dynamics. a** Communities identified from dynamic network analysis that transcend subunit divisions. The TRC–TFIIH–Ub complex is shown in anterior and posterior views with communities color-coded and labeled. **b** Graph of dynamic communication among communities. Nodes are sized by the number of residues in each community. Edge thickness represents the magnitude of dynamic communication between communities (betweenness). **c** Cluster of dynamic communities interacting strongly with CSB, CSA, and ELOF1; **d** Cluster of dynamic communities surrounding STK19; **e** Cluster of dynamic communities involving ubiquitin moieties Ub1, Ub2, and Ub3, CSB's UBD, the PHD of p62, CSA, ELOF1, STK19, and DDB1; **f** Labels identifying the domains or structural elements participating in each dynamic community. Network subgraphs for (**c**–**e**) are shown as insets.

precise CRL4$^{CSA}$ assembly. Here we extend these findings by showing that ELOF1 not only stabilizes CSA but also, by repositioning the VHS domain, arrests the rotation of the Cullin4A arm to allow access of the E2D2 enzyme to K1268 for specific ubiquitin transfer. Conversely, in the absence of ELOF1, the Cullin arm moves rapidly past the K1286 position to enable polyubiquitination of CSB. Thus, we posit the formation of TCR–CRL4$^{CSA}$ serves as a key decision point for the cell.

Successful K1268 ubiquitin transfer directs the cell to repair the DNA damage via the TCR pathway. Conversely, failure of TCR causes CSB and/or Pol II polyubiquitination and proteasomal degradation[53]. Furthermore, by modeling the entire TCR–CRL4$^{CSA}$ complex, including DDB1, Cullin4A, RBX1, the E2 ubiquitin ligase, Ub, NEDD8, and ELOF1, we uncover the functional dynamics and key structural determinants for efficient Pol II and UVSSA ubiquitination on which TCR depends.

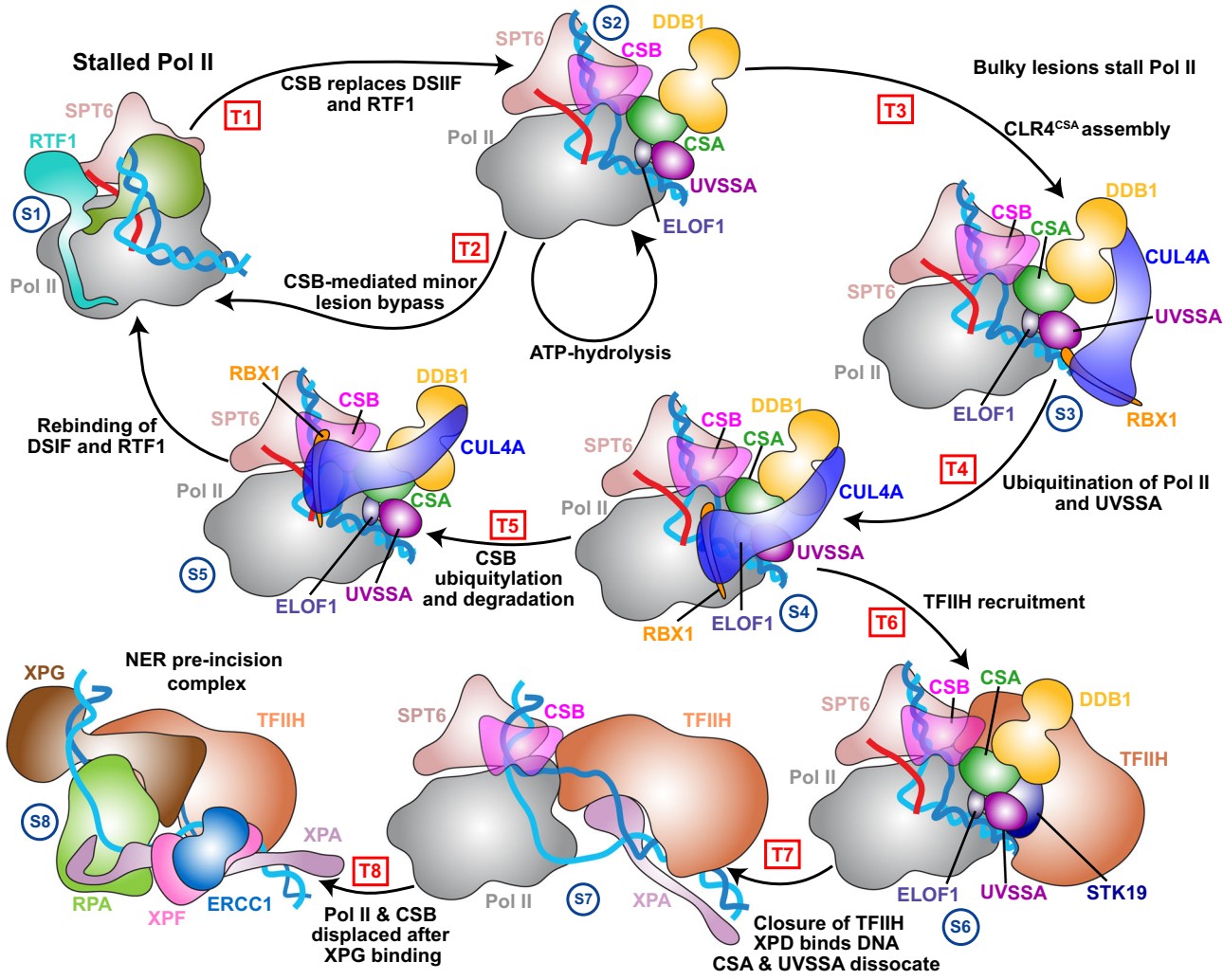

**Fig. 8 | Structure-based TCR mechanism from integrative computational modeling.** The schematic represents key steps in the TCR pathway –1) RNAPIIo lesion recognition and CSB recruitment; 2) CSB ATPase activity to bypass small lesions; 3) CLR4$^{CSA}$ assembly; 4) Pol II and UVSSA ubiquitination by CRL4$^{CSA}$; 5) CSB ubiquitylation and degradation to restore the transcription complex; 6) TFIIH recruitment to the nascent TCR complex; 7) XPB-unwinding activity expands the transcription bubble; XPA insertion at the downstream DNA junction, MAT1 departure and STK19 relocation trigger TFIIH closure and activation of XPD; 8) XPD pulling on the template strand to dislodge the lesion from Pol II; backtracking/displacement of Pol II by XPA and XPG leading to the formation of a NER pre-incision complex. Core TCR factors, RNAPIIo and TFIIH are shown in cartoon representation and color-coded. Blue arrows show the direction of DNA movement during the different stages of TCR. This figure was partially adapted from a model presented in a study by Kokic et al.[48]

Initial tethering of TFIIH to the TCR complex is driven by the flexible attachment of UVSSA's TIR to the p62 PH domain. This undoubtedly occurs in many possible orientations and necessitates interactions with additional protein partners to restrict TFIIH binding and enable DNA remodeling.

Our integrative model of the TCR–TFIIH assembly elucidates the functional association of TFIIH with a myriad of protein partners, including CSB, CSA, UVSSA, ELOF1, and the newly discovered critical-for-TCR protein STK19. Collectively, these factors facilitate TFIIH positioning onto the downstream DNA duplex in front of the lesion-arrested RNAPIIo. We find that STK19 forms multivalent protein interactions with CSA, Rpb1, and UVSSA's VHS domain, which allow STK19 to serve as a landing platform for the incoming TFIIH. Further interactions of STK19 with XPB, XPD, p62, and MAT1 orient TFIIH to allow precise engagement to the downstream DNA duplex. Moreover, we show that K1268 polyubiquitin chain formation and the UBD of CSB[54,55] strengthen the contacts of VHS with PHD and K414-ubiquitinated UVSSA, forming a bridge between RNAPIIo and TFIIH. Thus, our findings provide unexpected insight into the role of

ubiquitin modifications in activating TCR. We also note that the conformation adopted by TFIIH in the nascent TCR assembly is reminiscent of the open "horseshoe shaped" TFIIH in the PIC – an observation which poses new mechanistic questions. How could TFIIH in this open conformation possibly backtrack or displace RNAPIIo? How could DNA unwinding ahead of the transcription bubble expose the lesion to downstream repair factors? Clearly, a mechanism must exist to reshape the TCR machinery, allowing TFIIH to adopt a repair-competent conformation. To resolve this conundrum, we compared structural models of the TCR complex with open and circularized TFIIH as well as a recent integrative model of the GG-NER pre-incision complex[71] (Supplementary Fig. 9). Here we posit a mechanism wherein TFIIH first unwinds downstream dsDNA via its XPB subunit, expanding the transcription bubble in complete analogy to the PIC during transcription initiation. While such an expansion cannot directly dislodge the bulky lesion from the Pol II cleft, it eventually creates longer non-template ssDNA to enable XPA binding. Sufficiently long ssDNA provides a landing platform for XPA[73], which has affinity for the DNA junction near XPB (Fig. 8 and

Supplementary Fig. 9b). Binding of XPA and MAT1 to the TCR complex is mutually exclusive as the N-terminus of XPA outcompetes MAT1 for binding to the XPD Arch domain[70,72]. In turn, MAT1 displacement abolishes the interface of TFIIH with the Pol II stalk and may additionally destabilize STK19 by removing its contacts with MAT1's RING domain. Together, MAT1 removal and STK19 repositioning induce closure of TFIIH and enable XPD to bind the ssDNA of the template strand (Fig. 8 and Supplementary Fig. 9b). Moreover, STK19 displacement may facilitate XPD activation by removing the p62 lock on the XPD's ATPase domains. Additionally, we posit that TFIIH circularization allows XPD to pull concertedly on the template stand. Importantly, unlike XPB unwinding, XPD pulling occurs in the right direction to dislodge the DNA lesion from the polymerase cleft. At this stage, the Pol II-CSB-dsDNA interaction is the only significant hold that Pol II has on the TCR complex. Thus, eventual dissociation of Pol II from the evolving TCR machinery is triggered by XPG outcompeting CSB for binding to the upstream DNA duplex (Supplementary Fig. 9). As a final step in our proposed mechanism, the TCR complex disassembles and is replaced by a repair assembly similar to the pre-incision complex in GG-NER[71]. Intriguingly, superimposing the closed-TFIIH TCR complex onto the pre-incision complex (Supplementary Fig. 9d) shows that the XPG nuclease occupies a position partly overlapping with CSB in the TCR complex and is therefore poised to outcompete CSB for binding to the upstream DNA duplex. Thus, we hypothesize that, instead of Pol II backtracking, the combined association of XPA and XPG to the developing TCR machinery triggers the eventual dissociation of Pol II.

Collectively, these findings elucidate the structure and dynamics of critically important states of the TCR machinery. The practically complete TCR–TFIIH and TCR–CRL4$^{CSA}$ models and computational analyses yield key mechanistic insights into the assembly and regulation of early TCR intermediates, the structural basis of coordinated TFIIH recruitment, and the role of ubiquitin modifications in directing the cellular DNA damage response toward repair or proteasomal degradation.

Several experimentally determined Pol II–TCR–STK19 structures were published during the review of our paper[60–62]. The new cryo-EM structures were consistent with our AlphaFold-predicted interactions of STK19 with the Pol II Rpb1 clamp, CSA, and UVSSA. Accompanying TFIIH recruitment models were also published that involved TFIIH in its closed form and emphasized the STK19–XPD interaction identified by AlphaFold2. To achieve the posited binding mode, STK19 must relocate to the back side of XPD and let go of its binding interface with UVSSA. By contrast, the positioning of STK19 in our model does not require STK19 relocation and maintains existing UVSSA contacts. Thus, validation of the proposed TFIIH interaction interfaces would be of great interest as a topic of future research.

## Methods
### Model building
**DDB1 TCR subcomplexes.** We first built models for three TCR assemblies without TFIIH, TCR–DDB1, TCR–DDB1–VHS, and TCR–CSA–DDB1–VHS–ELOF1 based on cryo-EM structures 7OOB, 7OOP[48], and 8B3D, respectively (Supplementary Fig. 10). The placement of the mobile BPB domain of DDB1 in all three models reflected the 7OOP structure. Additionally, transcription elongation factor SPT6 and the mRNA accommodated in the Pol II exit tunnel were introduced into the TCR–DDB1 and TCR–DDB1–VHS models by superimposition onto the Rpb1 subunit of the 7OOP structure. In all models, the upstream and downstream DNA duplexes were extended to establish dsDNA contacts with SPT6 and the VHS domain of UVSSA. Additionally, for each model missing segments of the non-template DNA strand were built de novo to complete the transcription bubble. Any missing protein regions shorter than 20 amino acids were reconstructed using Modeller[85].

**TCR–CRL4$^{CSA}$ complexes.** To model the segmental flexibility of the DDB1 BPB domain, we built models of the TCR–CRL4$^{CSA}$ complex without ELOF1 in three distinct β-propeller conformations—linear, hinged, and twisted. The orientations of BPB in our models were based on the 4CI1, 5HXB, and 2HYE cryo-EM structures[74], respectively). The core of the TCR–CRL4$^{CSA}$ complex was based on the 7OPC cryo-EM structure[48], which did not have ELOF1 (Supplementary Fig. 10). Alignment of the distinct DDB1 conformers was done by superposition onto the BPA and BPC domains of the 7OPC structure. CUL4A and RBX1 were incorporated into the models by superposition of the BPB/CUL4A/RBX1 segment from 7OPC onto the linear, hinged, and twisted BPB conformer, respectively.

We also built models of the complete TCR–CRL4$^{CSA}$ in the presence of ELOF1 (Supplementary Fig. 10). In these models, CSA, CSB, and DDB1 were positioned according to the 8B3D cryo-EM structure[47], which had ELOF1 resolved. Across all TCR–CRL4$^{CSA}$ complexes, E2D2, NEDD8, and ubiquitin were modeled relative to CUL4A and were based on the 8B3I cryo-EM structure[47]. To model the ubiquitination-competent state of the TCR–CRL4$^{CSA}$ complex, we used a rotational conformer taken from the PNEB simulation trajectories. This conformation was observed along the computed MFEP and brought ubiquitin into closest proximity to the K1268 site, suggesting it was optimally positioned for efficient ubiquitination.

**TCR–TFIIH complex.** The integrative model of the TCR–TFIIH complex was based on the 8B3D cryo-EM structure[47] (suitably extended as in the TCR–DDB1–VHS–ELOF1 model) and the TFIIH conformation from the 6O9L cryo-EM structure of the human PIC[66] (Supplementary Fig. 10). STK19 was positioned in the TCR–TFIIH complex based on AlphaFold2 (ColabFold)[63,64] and AlphaFold3-multimer[65] predictions (Supplementary Fig. 11). First, we created STK19 complexes with CSA and the UVSSA VHS domain, and separately with the clamp head region of Rpb1 using AlphaFold2-multimer. We also confirmed that all three proteins, CSA, VHS, and Rpb1 clamp head could bind STK19, simultaneously. Superposition onto Rpb1 allowed placement of STK19 into the complete TCR–TFIIH structure. Aligning onto Rpb1 positioned STK19 next to XPB and the downstream DNA duplex. To capture DNA binding to STK19, we employed AlphaFold3, which unlike AlphaFold2 can predict protein–DNA interactions. The p62 PH domain bound to the TIR of UVSSA was built from the 5XV8 NMR structure[56]. AlphaFold2 calculations also predicted complex formation between the PHD–TIR and the UVSSA VHS domain, which allowed us to position PHD into the overall integrative model based on alignment to the VHS.

**TCR–TFIIH–Ub complex.** Our modeling took advantage of the previously built TCR–TFIIH model without ubiquitin. To incorporate ubiquitin moieties into TCR–TFIIH, we followed three different strategies. First, we used the UBQ_Gp_LYX-Cterm protocol[86] from the Rosetta modeling suite to introduce ubiquitin (Ub1) linked to the K1268 residue of the Pol II Rpb1 subunit. The protocol performs conjugated protein-protein docking, which samples the conformational ensemble available to the ubiquitin moiety (PDB ID: 1UBQ)[87] on the molecular surfaces of Rpb1, Rpb9, ELOF1 and UVSSA VHS while preserving the Ub-G76 linkage to Rpb1 K1268. A total of 10,000 docked conformers were generated, clustered by RMSD values, and the lowest energy (i.e., lowest Rosetta score) cluster centroid was selected and used for TCR–TFIIH–Ub model building. Second, we extended the polyubiquitin chain by a Ub2 monomer, which was positioned using an NMR-based structure of di-ubiquitin bound to the UBA domain of human Rad23A (PDB ID: 1ZO6)[80]. The CSB UBD domain was placed into our model based on superimposition onto UBA from 1ZO6. The third ubiquitin moiety (Ub3) was modeled from the 5XV8 structure[56] by AlphaFold2 prediction (Supplementary Fig. 11), which showed the formation of a complex of the p62 PHD domain with ubiquitin and the UVSSA VHS domain. We then conjugated Ub3 to the K414 of the UVSSA

TIR and incorporated the entire structural module into TCR–TFIIH–Ub by superimposing onto the VHS.

## Molecular dynamics

Molecular dynamics simulations of the TCR–TFIIH, TCR–TFIIH–Ub, and TCR–CRL4$^{CSA}$ complexes were performed using the Frontier supercomputer at the Oak Ridge Leadership Computing Facility. All systems (Supplementary Table 1) were set up with the TLeap module of AMBER[88], utilizing TIP3P water molecules for solvation[89]. Sodium counterions (Na$^+$) were added to neutralize the overall system charge, and additional Na$^+$ and Cl$^-$ ions were introduced to achieve a physiological salt concentration of 150 mM. All simulations were performed with NAMD3[90] using the Parm14SB[91] and OL15[92] AMBER force fields. Energy minimization was performed using NAMD for 5000 steps with positional restraints applied to the backbone atoms of all protein and nucleic acid chains. The systems were gradually heated to 300 K over 100 ps and equilibrated in the NVT ensemble. Positional restraints ($k = 10$ kcal mol$^{-1}$ Å$^{-2}$) were maintained on all heavy atoms of the protein-nucleic acid complex during this stage of the protocol. Equilibration continued for an additional 5 ns in the NPT ensemble, with gradual release of the positional restraints. Production simulations were carried out under NPT conditions (1 atm, 300 K) for 1 μs. The temperature was regulated using a Langevin thermostat with a coupling coefficient of 10 ps$^{-1}$, while the pressure was controlled using the Nosé-Hoover Langevin piston, set with an oscillation period of 200 fs. In all simulations, the first 50 ns of the production phase were allocated to ensure proper system equilibration. Therefore, the subsequent 950 ns of the production run were used for analysis. Long-range electrostatic interactions were computed using the particle mesh Ewald (PME) method. The r-RESPA multiple-time-step method[93] was employed with a 2-fs timestep for bonded and short-range non-bonded interactions and a 4-fs timestep for long-range electrostatics. Short-range non-bonded interactions were treated with a cutoff of 12 Å and a switching function beginning at 8.5 Å. Covalent bonds involving hydrogen atoms were constrained using the SHAKE algorithm. The resulting trajectories were analyzed using the CPPTRAJ module of AMBER and PyContact. Persistent contacts were identified using cutoff values of 3.5 Å for hydrogen bonds and salt-bridge interactions, and 4.5 Å for hydrophobic interactions. Only contacts observed in more than 60% of the trajectory frames were considered persistent. UCSF Chimera[94] was used for structure visualization and the packages COOT, Phenix, MDTraj, NetworkX, and PyContact were used for analysis.

## Partial nudged elastic band method

To explore the conformational dynamics of the TCR–CRL4$^{CSA}$ complex, we relied on the partial nudged elastic band (PNEB) method[75]. Using PNEB, we computed the minimum free energy path (MFEP) between the linear, hinged, and twisted conformations of TCR–CRL4$^{CSA}$. We first equilibrated the three conformational states in unbiased MD runs over 20 nanoseconds. From the equilibrated trajectories, we selected starting conformations to seed the PNEB path optimization. PNEB was performed with 60 replicas, which corresponded to conformational intermediates along the Cullin arm rotational transition. All heavy atoms of the protein-nucleic acid complexes were used in the definition of the NEB band. We gradually increased the temperature of the systems from 0 to 300 Kelvin using Langevin dynamics, setting at a collision frequency of 1000 ps$^{-1}$. PNEB spring forces of 10 kcal mol$^{-1}$ Å$^{-2}$ between adjacent replicas. The production runs were conducted at 300 K for 5 ns. Path convergence was monitored by the changes in protein and nucleic acid backbone atoms RMSD values computed between replicas.

## Covariance-based community network analysis

A covariance-based community network analysis[82,83] was conducted on the TCR–TFIIH–Ub MD trajectory ensemble. Residue-residue correlations were quantified using the CPPTRAJ module of AMBER[95]. First, a contact map was generated from the MD ensemble using the MDTraj package[96]. Network nodes corresponded to the Cα and P atoms of the protein or DNA residues, respectively. Network edges represented contacts between these residues. Two non-adjacent residues were considered to be in contact if they remained within 4.5 Å for at least 75% of the trajectory. The edges were weighted ($w_{i,j}$) according to the formula: $w_{i,j} = -\ln(|c_{i,j}|)$, where $c_{i,j}$ denotes the correlation coefficient between the residue pairs. To subdivide the protein network into dynamic communities, we employed the Girvan-Newman algorithm[97], which utilizes the betweenness centrality measure–a parameter calculated by the number of shortest paths crossing an edge, reflecting the likelihood of information transfer between nodes (residues). The algorithm iteratively identifies and removes the edge with the highest betweenness. This process of subdivision continues until a plateau is reached in the modularity score. Through this process, the TCR–TFIIH ensemble was partitioned into 36 distinct dynamic communities, yielding a high modularity score of 0.918. Finally, we computed the cumulative betweenness of inter-community edges to assess the strength of communication between dynamically correlated residue sets within the TCR–TFIIH complex. Protein network graphs were generated with Cytoscape. Convergence of the network analysis results was confirmed by carrying out the analysis on the first versus second half of trajectory frames (Supplementary Fig. 12).

## Reporting summary

Further information on research design is available in the Nature Portfolio Reporting Summary linked to this article.

## Data availability

Integrative models of the TCR complexes in this study have been deposited in the ModelArchive database with accession codes: 1) ma-blbmg [https://www.modelarchive.org/doi/10.5452/ma-blbmg] (TCR–TFIIH–Ub integrative model); 2) ma-k9i3k [https://www.modelarchive.org/doi/10.5452/ma-k9i3k] (TCR–CRL4$^{CSA}$ complex in the presence of ELOF1); 3) ma-89py8 [https://www.modelarchive.org/doi/10.5452/ma-89py8] (TCR–CRL4$^{CSA}$ complex without ELOF1 in the twisted BPB conformation); 4) ma-6cl90 [https://www.modelarchive.org/doi/10.5452/ma-6cl90] (TCR–CRL4$^{CSA}$ complex without ELOF1 in the hinged BPB conformation); 5) ma-pdzpq [https://www.modelarchive.org/doi/10.5452/ma-pdzpq] (TCR–CRL4$^{CSA}$ complex without ELOF1 in the linear BPB conformation). PDB accession codes used in the study: 1) 8B3D; 8B3I; 1UBQ; 1ZO6 [https://www.modelarchive.org/doi/10.5452/ma-cahen]. Initial and final configurations from the MD trajectories are included in Supplementary Data 1.

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

## Acknowledgements

We thank Professor Jun Yin (Georgia State University) for insightful discussions. This work was supported by the National Institute of General Medical Sciences grant R35GM139382 (I.I.), the National Institute of Environmental Health Sciences grant R01 ES032786 (I.I., S.E.T.), the National Science Foundation grant MCB–2027902 (I.I.), National Institute of General Medical Sciences grant R01GM102362 (D.W.), and National Cancer Institute grants P01 CA092584 (J.A.T., I.I., S.E.T.), R35 CA220430 (J.A.T.). An award of computer time to I.I. was provided by the INCITE program. This research also used resources of the Oak Ridge Leadership Computing Facility, which is a DOE Office of Science User Facility supported under Contract DE-AC05-00OR22725.

## Author contributions

I.I. directed the study. T.P., C.Y., and I.I. contributed to the design of the study. T.P., C.Y. performed model building and molecular simulations of the models. T.P., C.Y., J.Y., S.E.T., J.A.T., D.W., and I.I. analyzed the data. T.P., C.Y., S.E.T., J.A.T., D.W., and I.I. wrote the manuscript.

## Competing interests

The authors declare no competing interests.
