## [Transparent Peer Review file · Nature Communications]

Molecular model of TFIIH recruitment to the transcription-coupled repair machinery

Corresponding Author: Professor Ivaylo Ivanov

Version 0:

Reviewer comments:

Reviewer #1

(Remarks to the Author)

The manuscript by Paul et al. presents a comprehensive study that leverages existing structural data in conjunction with AlphaFold predictions to construct an integrative model of the initial TFIIH-bound transcription-coupled repair (TCR) complex. TCR complexes are highly dynamic, complex protein assemblies that undergo constantly assembly and disassembly processes to achieve function. In this work, the authors draw on their understanding of TCR processes and structural data to construct three multi-protein complexes: the TCR–TFIIH complex, TCR–CRL4CSA complex, and TCR–TFIIH–Ub complex. These models yield critical mechanistic insights into the assembly and regulation of early TCR intermediates, including the structural underpinnings of TFIIH recruitment coordination and the role of ubiquitin modifications in directing DNA repair or proteasomal degradation.

The successful modeling of these large, dynamic, multicomponent complexes using computational methods is a significant achievement that would be unattainable with experimental approaches alone. While direct experimental validation of these predicted structures is not provided, the authors have carefully contextualized their models within the framework of existing experimental data. Additionally, their integrative models offer a valuable roadmap for future experiments aimed at elucidating the mechanisms by which TCR machinery facilitates precise lesion repair and genome integrity preservation. Overall, this is a well-conceived, methodologically rigorous, and well-written study.

I have a few suggestions for the authors to consider:

1. While the authors provide indirect validation by comparison to existing experimental data, it would be beneficial to explore alternative approaches for quantifying model quality and uncertainty. Incorporating such quality control measures would significantly enhance the robustness and impact of the study's conclusions.

2. The authors appear to use AlphaFold2-multimer to predict protein-protein interaction interfaces, subsequently fitting these predicted dimers into larger complex structures. It would be valuable to discuss the uncertainties associated with this fitting approach, especially in light of potential cooperative binding or binding-induced conformational changes that may influence binding of another protein(s). Additionally, the authors could consider whether direct multimer prediction could be applied to some cases presented in this study, as this approach may better capture the dynamic and potentially weak interactions within these complexes.

3. Given the considerable size and complexity of these TCR complexes, it would be helpful to clarify how the authors determined the simulation duration necessary to capture functionally relevant dynamics. Additionally, information on how simulation convergence was assessed would strengthen the analysis. The authors state that the simulations reveal the functional dynamics of these systems; however, this may be an overstatement, given that (1) the models are computationally constructed and require relaxation time, and (2) functionally relevant dynamics in these complexes may occur on timescales likely exceeding 1-2 microseconds.

Reviewer #2

(Remarks to the Author)

I attached the review as a word document - I included a few images that cannot be pasted here. Hope that is ok.

Version 1:

Reviewer comments:

Reviewer #1

(Remarks to the Author)

My concerns have been satisfactorily addressed in the revisions. I have no further comments or concerns.

Reviewer #2

(Remarks to the Author)

I thank the reviews for answering my questions and comments. I have no further concerns.

Point by point response to reviewer comments:

Reviewer #1 (Remarks to the Author):

The manuscript by Paul et al. presents a comprehensive study that leverages existing structural data in conjunction with AlphaFold predictions to construct an integrative model of the initial TFIIH-bound transcription-coupled repair (TCR) complex. TCR complexes are highly dynamic, complex protein assemblies that undergo constantly assembly and disassembly processes to achieve function. In this work, the authors draw on their understanding of TCR processes and structural data to construct three multi-protein complexes: the TCR–TFIIH complex, TCR–CRL4CSA complex, and TCR–TFIIH–Ub complex. These models yield critical mechanistic insights into the assembly and regulation of early TCR intermediates, including the structural underpinnings of TFIIH recruitment coordination and the role of ubiquitin modifications in directing DNA repair or proteasomal degradation.

The successful modeling of these large, dynamic, multicomponent complexes using computational methods is a significant achievement that would be unattainable with experimental approaches alone. While direct experimental validation of these predicted structures is not provided, the authors have carefully contextualized their models within the framework of existing experimental data. Additionally, their integrative models offer a valuable roadmap for future experiments aimed at elucidating the mechanisms by which TCR machinery facilitates precise lesion repair and genome integrity preservation. Overall, this is a well-conceived, methodologically rigorous, and well-written study.

Thank you.

I have a few suggestions for the authors to consider:

1. While the authors provide indirect validation by comparison to existing experimental data, it would be beneficial to explore alternative approaches for quantifying model quality and uncertainty. Incorporating such quality control measures would significantly enhance the robustness and impact of the study's conclusions.

We agree with the reviewer that independent measures of model quality could substantively improve the impact of the manuscript. Thus, besides evaluating AlphaFold predictions based on pLDDT scores, we have run additional analyses.

We followed the reviewer's suggestion and examined the quality of the interfaces in the TCR-TFIIH and CLR4^{CSA} models.

1. The deposited TCR-TFIIH and CLR4^{CSA} models have undergone real space refinement with the Phenix package followed by local refinement with Coot. We did not observe any obvious issues with stereochemistry, clashes, or significant molecular geometry violations for the interfacial residues (see validation tables below and a screenshot from the Coot session; green bars along the protein chains indicate the geometry is OK for the particular residues shown):

Validation statistics after real space refinement	TCR-TFIIH model
MolProbity score	1.93

MolProbity Clashscore	10.86
Rotamers outliers (%)	1.40
C β deviations (%)	0.02
Ramachandran favored (%)	96.11
Ramachandran allowed (%)	3.76
Ramachandran outliers (%)	0.13

Validation statistics after real space refinement	CLR4 ^{CSA} model
MolProbity score	1.97
MolProbity Clashscore	10.43
Rotamers outliers (%)	1.50
C β deviations (%)	0.05
Ramachandran favored (%)	95.67
Ramachandran allowed (%)	4.20
Ramachandran outliers (%)	0.13

Sample screenshots from Coot showing the model quality are included below:

TCR-TFIIH model:

CLR4^{CSA} model:

2. We performed interface analysis for the key STK19 and TFIIH interfaces with the PISA server. Results are summarized below.

Interface	STK19		XPB	
Number of atoms	135	7.9%	131	2.5%
Number of residues	39	17.8%	37	5.7%
Interface surface area, Å ²	1129.6			

Interface	STK19		XPB	
Number of atoms	21	1.2%	22	0.4%
Number of residues	10	4.6%	7	0.9%
Interface surface area, Å ²	196.4			

gain on complex formation, kcal/mol	-14.3	
N _{HB}	7	
N _{SB}	3	

gain on complex formation, kcal/mol	0.6	
N _{HB}	5	
N _{SB}	0	

Interface	STK19		P62	
Number of atoms	87	5.1%	87	2.0%
Number of residues	29	13.2%	29	5.2%
Interface surface area, Å ²	829.8			
gain on complex formation, kcal/mol	-5.3			
N _{HB}	9			
N _{SB}	4			

Interface	STK19		MAT1	
Number of atoms	54	3.1%	44	2.4%
Number of residues	13	5.9%	14	6.4%
Interface surface area, Å ²	444.7			
gain on complex formation, kcal/mol	0.1			
N _{HB}	12			
N _{SB}	5			

Interface	STK19		DNA	
Number of atoms	46	2.7%	33	2.0%
Number of residues	15	6.9%	7	8.8%
Interface surface area, Å ²	317.3			
gain on complex formation, kcal/mol	-5.9			
N _{HB}	1			
N _{SB}	4			

Interface	UVSSA		P62	
Number of atoms	61	2.6%	76	1.7%
Number of residues	13	4.5%	21	3.8%
Interface surface area, Å ²	608.5			
gain on complex formation, kcal/mol	-3.8			
N _{HB}	9			
N _{SB}	4			

Interface	UBD(CSB)		UB1	
Number of atoms	49	0.9%	48	8.0%
Number of residues	13	1.9%	18	23.7%
Interface surface area, Å ²	434.4			
gain on complex formation, kcal/mol	-2.6			
N _{HB}	5			
N _{SB}	4			

Interface	PHD (P62)		UB3	
Number of atoms	55	1.2%	48	8.0%
Number of residues	13	2.3%	15	19.7%
Interface surface area, Å ²	487.8			
gain on complex formation, kcal/mol	-3.2			
N _{HB}	6			
N _{SB}	3			

Interface	PHD (P62)		VHS (UVSSA)	
Number of atoms	76	1.7%	61	2.6%
Number of residues	21	3.8%	13	4.5%
Interface surface area, Å ²	608.5			
gain on complex formation, kcal/mol	-3.8			
N _{HB}	9			
N _{SB}	4			

** N_{HB} = Number of hydrogen bonds

** N_{SB} = Number of salt bridges

2. The authors appear to use AlphaFold2-multimer to predict protein-protein interaction interfaces, subsequently fitting these predicted dimers into larger complex structures. It would be valuable to discuss the uncertainties associated with this fitting approach, especially in light of potential cooperative binding or binding-induced conformational changes that may influence binding of another protein(s). Additionally, the authors could consider whether direct multimer prediction could be applied to some cases presented in this study, as this approach may better

capture the dynamic and potentially weak interactions within these complexes.

We agree that the sequential recruitment of TCR factors may involve binding-induced conformational changes. This potential concern is mitigated in part by the microsecond timescale MD simulations used to relax the assemblies. These simulations optimize the AlphaFold predicted interfaces to the extent possible. Moreover, we did not limit ourselves to binary complex prediction with AF2. Instead, we employed AlphaFold3 and performed multimer predictions that included DNA and as many of the STK19- and MAT1-interacting protein chains as was practical (see Supplementary Figure 11b in the revised manuscript). The AlphaFold3 prediction encompassed 4,649 amino acid residues, close to the upper limit of 5000 tokens that can be handled by the server. Protein chains distal from the CSB/CSA/UVSSA and TFIIH interfaces were modeled based on the available cryo-EM structures rather than AlphaFold.

3. Given the considerable size and complexity of these TCR complexes, it would be helpful to clarify how the authors determined the simulation duration necessary to capture functionally relevant dynamics. Additionally, information on how simulation convergence was assessed would strengthen the analysis. The authors state that the simulations reveal the functional dynamics of these systems; however, this may be an overstatement, given that (1) the models are computationally constructed and require relaxation time, and (2) functionally relevant dynamics in these complexes may occur on timescales likely exceeding 1-2 microseconds.

To monitor the relaxation dynamics of the complexes, we relied on the change in the RMSD from the initial model (computed over C α in and P atoms). We monitored RMSD convergence and excluded from analysis the first ~50 ns of trajectory frames. The plot below illustrates the backbone RMSD change during the first 85 ns of simulation for one of the independent trajectories.

Generally, 1-2 microseconds are sufficient to obtain converged covariance matrices for network analysis. Indeed, network graphs and dynamic community splittings showed little change if analyzed over a subset (one half) of trajectory frames.

Network graphs of the TCR-TFIIH complex obtained by averaging over a) the first or b) the second half of trajectory frames.

By ‘functional dynamics’ we mean functionally relevant motions on timescales that are accessible to us via MD simulations. We have now made that distinction explicit on page 17 of the revised manuscript: “ ... and analyzed its functional dynamics on timescales accessible to MD simulations.”

Undoubtedly, there are other functionally relevant motions and dynamic conformational changes that occur on much longer timescales (milliseconds to seconds). However, such motions are beyond the scope of the present manuscript.

Reviewer #2 (Remarks to the Author):

The authors have used existing structural data and AlphaFold multimer to model integration of STK19 and TFIIH into a growing TCR assembly. Simulations were applied to the model to obtain a more dynamic insight into the workings of the repair machinery, including the activity of TFIIH and ubiquitylation of Pol II.

The central novel point of this work is characterisation of STK19 binding at the position suitable for TFIIH recruitment. Even though this insight showcases the power of molecular modelling with AlphaFold, experimentally determined PolII-TCR-STK19 structures with accompanying AlphaFold models of TFIIH recruitment were recently published by several different labs.

We appreciate the reviewer’s favorable comments regarding the power of modeling with AlphaFold and the novel role of STK19 binding in TFIIH recruitment. As the reviewer aptly pointed out, several new STK19-related papers have appeared since submission. We have cited them in our revised manuscript.

Major comments:

1. *The title suggests the mechanism was experimentally determined. Appropriate title would be “The molecular model of TFIIH..”, otherwise very misleading.*

We have modified the title as suggested to emphasize that this is a molecular modeling study: “Molecular model of TFIIH recruitment to the transcription-coupled repair machinery.”

2. *The positioning of TFIIH seems to be very biased by the positioning of TFIIH in the PIC – the model was made by adding TFIIH from PIC onto Pol II-TCR structure, so no novel TFIIH conformation was considered. Authors claim that majority of TFIIH- STK19*

interface is between XPB and STK19. Yet, the interface was not particularly well shown and no comments on the confidence of predicting this interaction were made. In the Sup. Fig. 10 where the confidence of predictions is shown, this interface is not presented in detail, it is contained in panel B but cannot really be inspected. Author should clearly comment of the confidence of prediction for all major interaction interfaces mentioned in the manuscript, and STK19-XPB interface should be especially rigorously analysed and presented. Since the manuscript is based of AlphaFold modelling, accuracy of prediction should be more discussed and more clearly visualized overall. Also, for example, how confident is p62-STK19 interface prediction, very hard to say from looking at Fig. 3i. What about the contact to the anchor loop (not clearly labelled in the figure)?

We appreciate the reviewer's suggestions and have added two new panels to Supplementary Fig. 11 to show the predicted STK19-XPB (Fig. S10.c) and UVSSA-XPB (Fig. S10.d) interfaces more clearly. Please also see our response to the first point made by Reviewer #1. We have also added a new Supplementary Fig. 5, which analyzes the detailed contacts between XPB, p62 and STK19.

3. Last figure is very similar to already published figure from DOI: [10.1038/s41586-021-03906-4](https://doi.org/10.1038/s41586-021-03906-4), outline is the same and most of the text is even copy-pasted. It should -at least- be mentioned in the figure legend this figure was created based on the previously published one. I placed the two figures here for comparison:

Fig. 8 was partially adapted from a model presented in Figure 4 of the Kokic et. al study (Nature 598, 368–372 (2021)). This was indeed a very influential paper on the structural basis of transcription–TC-NER coupling. We thought it important to present our new findings in the context of the established model. We cite the Kokic study as reference #48 in our manuscript and we have added a clarification to the figure legend: “... partially adapted from a figure in Kokic et. al.[48]”

New Figure 8

Yet, there are important distinctions between the two figures and the respective models. For example, state S5 in our model is a branching point, which directs the cell toward DNA repair or alternatively toward CSB ubiquitination and degradation. Correspondingly, state S7 is not part of the cycle but constitutes the first in a series of states forming the TC-NER pathway.

4. Authors propose a model of how TFIH works to expose the lesion for DNA repair. From their model I still do not understand how TFIH initially unwinds the DNA and exposes it

sufficiently for XPA to bind or for XPD to be deposited onto the template strand DNA. It is also unclear how Pol II would be moved away.

We made changes to the Discussion section (on pages 18 and 19) to explain these points more clearly. Briefly, XPB unwinds the dsDNA exactly as it would in the PIC, expanding both the template and non-template ssDNA strands downstream of the lesion. Such expansion cannot directly dislodge the lesion, but it provides sufficient ssDNA footprint on the non-template strand for XPA to engage the downstream DNA junction. Binding of XPA and MAT1 is mutually exclusive as the N-terminus of XPA outcompetes MAT1 for binding to the XPD Arch domain. Displacement of MAT1 abolishes the TFIIH interface with the Pol II stalk and additionally destabilizes STK19. These changes trigger TFIIH closure, which allows XPD to engage the expanded template-strand ssDNA as shown in state S8. In this orientation, XPD is positioned to pull the template strand in a direction that would dislodge the lesion (i.e., away from the Pol II active site). The eventual removal of Pol II is triggered by XPG outcompeting CSB for binding to the upstream duplex. At this stage (S8), the Pol II-CSB-dsDNA interaction is the only significant hold that Pol II has on the TCR complex. We also updated the legend of Fig. 8 to make the steps of the mechanism clearer.

5. *In light of new STK19 structures, additional paragraph should be included comparing the experimental findings and additional models, with predication in this work.*

Thank you for this suggestion. We have added a paragraph (on page 19) referring to the new experimental papers that have appeared since our manuscript submission. We also provide a brief comparison to our work.

Minor comments:

1. *“Yet no structural models exist of these factors, including CSA, UVSSA, and transcription factor IIIH (TFIIH), assembled into a functional repair complex.” is a bit misleading. Better would be – structure of Pol II bound to core TCR machinery was solved, but how TFIIH is recruited.*

We have changed the text of the abstract as suggested: “Yet it remains unknown how transcription factor IIIH (TFIIH) is recruited to the intact TCR complex.”

2. *“TFIIH harbors two translocase subunits XPB and XPD, which unwind DNA near the Pol II cleft and expose the DNA lesion” – has this been show? Ref?*

The DNA remodeling activities of XPB and XPD are required for functional NER. doi: 10.1146/annurev-biochem-052621-091205, Compe E, Egly J-M. 2016. Nucleotide excision repair and transcriptional regulation: TFIIH and beyond. Annu. Rev. Biochem. 85(1):265–90 (XPD catalytic activity required). Furthermore, biochemical evidence using a minimal TCR system suggests that the 3' incision by XPG, which is otherwise blocked by bound RNAPII, requires the ATP-dependent activity of TFIIH to displace RNAPII and gain access to the DNA junction (doi:10.1016/j.tcb.2021.02.007 & doi:10.1016/j.molcel.2005.09.022).

3. *“As the centerpiece of this complex TCR machinery, TFIIH unwinds DNA to expose the lesion, causes Pol II backtracking and/or dissociation, and coordinates the dual incision process. “ Has this been shown? Ref?*

Pol II backtracking vs. dissociation in TCR has been a matter of considerable debate (doi: 10.1146/annurev-biochem-052621-091205) with multiple mechanisms proposed. These mechanisms are not necessarily mutually exclusive and range from TFIIIS-induced Pol II backtracking to TFIIH-mediated Pol II dissociation and full or partial (Rpb1) Pol II degradation. While we favor the Pol II dissociation proposal (see discussion on page 18-19), we cannot exclude the other mechanisms. We have changed the text to make it less categorical: “As the centerpiece of this complex TCR machinery, TFIIH unwinds DNA to expose the lesion^{9, 46, 25}, **is considered to cause** Pol II backtracking and/or dissociation¹⁰, and coordinates the dual incision process.”

4. *Almost the entire section 317-354 and Fig.4 explain how ELOF1 stabilizes TCR assembly and restricts the CSA mobility (presented as novel data revealed by simulation), which was already experimentally shown and described by EM (DOI: [10.1038/s41594-023-01207-0](https://doi.org/10.1038/s41594-023-01207-0)).*

The effect of ELOF1 on CSA mobility was shown by MD simulations and also completely independently by EM. We have cited the EM study (reference #47) and noted its relevance.

Sup. Fig. 7. The portrayal of the complex makes the figure very difficult to read. In several places structures are shown in a flat 2D way without visible black outline, which makes visual inspection difficult (also in the movies).

We agree that Fig. S8 is difficult to follow but if we were to introduce depth cueing to give it a more 3D look that would interfere with community colors. Instead, we have changed movie 3 to enhance the 3D view of the communities. The movie will complement Fig. S8.

5. *Why isn't STK19 WH2 shown in Figure 1c? That seems to be the novelty.*

Initially, we debated whether to highlight STK19 or not in Fig. 1. However, the way the paper is structured, discussion of STK19 follows the description of the overall complex. We also wanted to avoid repetition with panels a, and g, in Fig. 3.

6. *Authors dismiss STK19-XPB interactions due to multivalent interactions between STK19 and non-TFIIH elements (253-253). That is a bit flawed because the novel factor here is TFIIH positioning, not STK19 binding to the other proteins. Authors should rather compare how extensive and confident are the interfaces between STK19 and XPB/XPB.*

To achieve the binding mode predicted by AlphaFold2, STK19 must relocate to the back side of XPB and let go of its binding interface with UVSSA (which is seen in cryo-EM). By contrast, STK19 binding to XPB and dsDNA does not require relocation and maintains existing UVSSA contacts.

7. *286-288 is written as if XPB recruits STK19 and not the other way around.*

“In this orientation, XPB is poised to facilitate STK19 association with duplex DNA extending from the XPB DNA-binding cleft.” We have changed the sentence to: “In this

orientation, STK19 associates with both XPB and the duplex DNA extending from the XPB DNA-binding cleft.”

8. 294 -*how does 140 degrees kink facilitate unwinding by XPB? How is DNA “guided” in PIC?*

The Rpb1 clamp, STK19 and XPB come together to form a continuous groove with favorable electrostatics that buttresses dsDNA on its path to the Pol II cleft. We suggest that this structural feature may facilitate unwinding by preventing transient DNA dissociation from XPB. There is no comparable ‘guiding’ feature in the PIC.

The authors have used existing structural data and AlphaFold multimer to model integration of STK19 and TFIIH into a growing TCR assembly. Simulations were applied to the model to obtain a more dynamic insight into the workings of the repair machinery, including the activity of TFIIH and ubiquitylation of Pol II.

The central novel point of this work is characterisation of STK19 binding at the position suitable for TFIIH recruitment. Even though this insight showcases the power of molecular modelling with AlphaFold, experimentally determined PolII-TCR-STK19 structures with accompanying AlphaFold models of TFIIH recruitment were recently published by several different labs.

Major comments:

1. The title suggests the mechanism was experimentally determined. Appropriate title would be the “The molecular model of TFIIH..”, otherwise very misleading.
2. The positioning of TFIIH seems to be very biased by the positioning of TFIIH in the PIC – the model was made by adding TFIIH from PIC onto Pol II-TCR structure, so no novel TFIIH conformation was considered. Authors claim that majority of TFIIH-STK19 interface is between XPB and STK19. Yet, the interface was not particularly well shown and no comments on the confidence of predicting this interaction were made. In the Sup. Fig. 10 where the confidence of predictions is shown, this interface is not presented in detail, it is contained in panel B but cannot really be inspected. Author should clearly comment of the confidence of prediction for all major interaction interfaces mentioned in the manuscript, and STK19-XPB interface should be especially rigorously analysed and presented. Since the manuscript is based of AlphaFold modelling, accuracy of prediction should be more discussed and more clearly visualized overall. Also, for example, how confident is p62-STK19 interface prediction, very hard to say from looking at Fig. 3i. What about the contact to the anchor loop (not clearly labelled in the figure)?
3. Last figure is very similar to already published figure from DOI: [10.1038/s41586-021-03906-4](https://doi.org/10.1038/s41586-021-03906-4), outline is the same and most of the text is even copy-pasted. It should -at least- be mentioned in the figure legend this figure was created based on the previously published one. I placed the two figures here for comparison:

4. Authors propose a model of how TFIIF works to expose the lesion for DNA repair. From their model I still do not understand how TFIIF initially unwinds the DNA and exposes it sufficiently for XPA to bind or for XPD to be deposited onto the template strand DNA. It is also unclear how Pol II would be moved away.
5. In light of new STK19 structures, additional paragraph should be included comparing the experimental findings and additional models, with predications in this work.

Minor comments:

1. “Yet no structural models exist of these factors, including CSA, UVSSA, and transcription factor IIF (TFIIF), assembled into a functional repair complex.” is a bit misleading. Better would be – structure of Pol II bound to core TCR machinery was solved, but how TFIIF is recruited..
2. “TFIIF harbors two translocase subunits XPB and XPD, which unwind DNA near the Pol II cleft and expose the DNA lesion” – has this been show? Ref?
3. “As the centerpiece of this complex TCR machinery, TFIIF unwinds DNA to expose the lesion, causes Pol II backtracking and/or dissociation, and coordinates the dual incision process. “ Has this been shown? Ref?
4. Almost the entire section 317-354 and Fig.4 explain how ELOF1 stabilizes TCR assembly and restricts the CSA mobility (presented as novel data revealed by simulation), which was already experimentally shown and described by EM (DOI: [10.1038/s41594-023-01207-0](https://doi.org/10.1038/s41594-023-01207-0)).
5. Sup. Fig. 7. The portrayal of the complex makes the figure very difficult to read. In several places structures are shown in a flat 2D way without visible black outline, which makes visual inspection difficult (also in the movies).
6. Why isn't STK19 WH2 shown in Figure 1c? That seems to be the novelty.
7. Authors dismiss STK19-XPD interactions due to multivalent interactions between STK19 and non-TFIIF elements (253-253). That is a bit flawed because the novel factor here is TFIIF positioning, not STK19 binding to the other proteins. Authors should rather compare how extensive and confident are the interfaces between STK19 and XPD/XPB.
8. 286-288 is written as if XPB recruits STK19 and not the other way around.
9. 294 -how does 140 degrees kink facilitate unwinding by XPB? How is DNA “guided” in PIC?